# Dynamic anticipation by Cdk2/Cyclin A-bound p27 mediates signal integration in cell cycle regulation

Maksym Tsytlonok[1], Hugo Sanabria [2,3], Yuefeng Wang[4,8], Suren Felekyan[3], Katherina Hemmen[3], Aaron H. Phillips[4], Mi-Kyung Yun[4], M. Brett Waddell[4,5], Cheon-Gil Park[4], Sivaraja Vaithiyalingam[4,5], Luigi Iconaru[4], Stephen W. White [4], Peter Tompa[1,6], Claus A.M. Seidel [3] & Richard Kriwacki [4,7]

p27[Kip1] is an intrinsically disordered protein (IDP) that inhibits cyclin-dependent kinase (Cdk)/cyclin complexes (e.g., Cdk2/cyclin A), causing cell cycle arrest. Cell division progresses when stably Cdk2/cyclin A-bound p27 is phosphorylated on one or two structurally occluded tyrosine residues and a distal threonine residue (T187), triggering degradation of p27. Here, using an integrated biophysical approach, we show that Cdk2/cyclin A-bound p27 samples lowly-populated conformations that provide access to the non-receptor tyrosine kinases, BCR-ABL and Src, which phosphorylate Y88 or Y88 and Y74, respectively, thereby promoting intra-assembly phosphorylation (of p27) on distal T187. Even when tightly bound to Cdk2/cyclin A, intrinsic flexibility enables p27 to integrate and process signaling inputs, and generate outputs including altered Cdk2 activity, p27 stability, and, ultimately, cell cycle progression. Intrinsic dynamics within multi-component assemblies may be a general mechanism of signaling by regulatory IDPs, which can be subverted in human disease.

[1] VIB Center for Structural Biology, Vrije Universiteit Brussel, Pleinlaan 2 1050 Brussels, Belgium. [2] Department of Physics and Astronomy, Clemson University, Clemson, SC 29634, USA. [3] Lehrstuhl für Molekulare Physikalische Chemie, Heinrich-Heine-Universität, 40225 Düsseldorf, Germany. [4] Department of Structural Biology, St. Jude Children's Research Hospital, 262 Danny Thomas Place, Memphis, TN 38105, USA. [5] Molecular Interaction Analysis Shared Resource, St. Jude Children's Research Hospital, 262 Danny Thomas Place, Memphis, TN 38103, USA. [6] Institute of Enzymology, Research Centre for Natural Sciences of the Hungarian Academy of Sciences, Budapest, Hungary. [7] Department of Microbiology, Immunology and Biochemistry, University of Tennessee Health Sciences Center, Memphis, TN 38163, USA. [8] Present address: Department of Radiation Oncology, West Cancer Center and Research Institute, Memphis, TN 38138, USA. These authors contributed equally: Maksym Tsytlonok, Hugo Sanabria, Yuefeng Wang. These authors jointly supervised this work: Peter Tompa, Claus A. M. Seidel, Richard Kriwacki. Correspondence and requests for materials should be addressed to P.T. (email: tompa@enzim.hu) or to C.A.M.S. (email: cseidel@hhu.de) or to R.K. (email: richard.kriwacki@stjude.org)

Many eukaryotic proteins lack stable 3D structure and exhibit conformational heterogeneity in isolation but fold into discrete conformations upon binding to other molecules (e.g., proteins, nucleic acids, small molecule ligands, metal ions, etc.)[1]. These so-called intrinsically disordered proteins (IDPs) often function to regulate complex biological processes[2], with their bound conformations exerting a basal level of regulatory control over their targets. However, this basal level of control can often be modulated, or switched, through post-translational modifications to achieve complex regulatory behavior[3,4], as exemplified by the cell-cycle regulator, p27$^{Kip1}$ (p27), which folds upon binding to cyclin-dependent kinase (Cdk)/cyclin complexes that control cell division[5]. The basal function of p27 is the inhibition of the kinase activity of Cdk/cyclin complexes, which can be relieved through phosphorylation of its tyrosine residue(s) by non-receptor tyrosine kinases (NRTKs)[6,7]. Relief of Cdk inhibition triggers a sequence of additional post-translational modifications, including phosphorylation of T187 to activate a phosphodegron within the same p27 molecule, followed by its ubiquitination and degradation, triggering full activation of Cdk/cyclin complexes and progression to S phase during cell division.

The existence of regulatory modification sites within bound and folded regions of p27 (and IDPs in general) raises a key question regarding how these sites become accessible for enzymatic modification in signal transduction. It has been previously suggested that regions of bound disordered proteins experience dynamic fluctuations[6,8]. Therefore, we hypothesized that regions within IDPs subject to function-altering post-translational modifications have evolved to dynamically sample different conformational states that provide accessibility to modifying enzymes. The ensuing enzymatic modifications remodel the conformational landscape of the IDP, enabling further changes and controlled integration of incoming signals. Herein, we tested this hypothesis by studying p27 bound to Cdk2/cyclin A by NMR spectroscopy, single-molecule multiparameter fluorescence detection (smMFD) and other biophysical techniques, which revealed dynamic fluctuations, which we refer to as dynamic anticipation. These allow for phosphorylation of two of its tyrosine residues, Y88[6] and Y74[7], by the non-receptor tyrosine kinases (NRTKs), BCR-ABL and Src, which in turn lead to potentiated downstream signaling through the Cdk2/cyclin A axis. We suggest that remodeling of dynamic structural ensembles of bound states by post-translational modifications might be a general mechanism of signal integration by regulatory IDPs.

## Results

**p27 phosphorylation restores Cdk2 activity.** p27 is tethered to the Cdk2/cyclin A assembly via two discontinuous subdomains, D1 and D2, within its kinase inhibitory domain (KID); D1 binds to a hydrophobic surface patch on cyclin A and D2 binds to the active site of Cdk2 (for subdomain nomenclature, see Fig. 1a, for the structure of the assembly, see Fig. 1b). Phosphorylation of Y88 by BCR-ABL was previously shown to partially relieve p27-dependent inhibition of Cdk2/cyclin A, triggering p27 ubiquitination and degradation, and full activation of Cdk2 to drive progression into S phase of the cell division cycle[6,7]. Tyrosine 88, located within D2 subdomain, is inserted into the ATP-binding pocket of Cdk2 (Fig. 1b)[5,6]. Interestingly, p27 exhibits a second tyrosine within its KID, Y74 (Fig. 1), and both Y88 and Y74 were previously shown to be phosphorylated by Src in a large fraction of hyper-proliferative breast cancer cell lines[7]. Dual phosphorylation of Y88 and Y74 was shown to be associated with heightened Cdk2 activity[6,7] but the molecular mechanism of this effect was not investigated. In Cdk2 activity assays with p27-KID, in

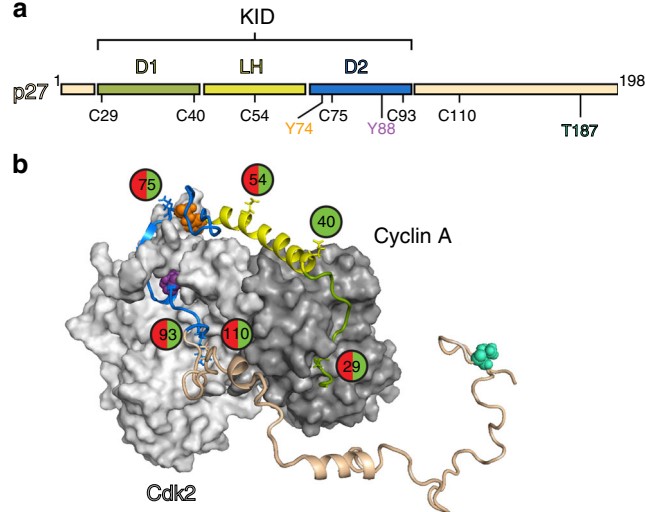

**Fig. 1** Domain structure of p27. **a** Full-length p27 contains the kinase inhibitory domain (KID), which is subdivided into domain 1 (D1), linker helix (LH), and domain 2 (D2). Tyrosine phosphorylation sites are labeled as Y74 and Y88 and the threonine phosphorylation site is labeled as T187. The labeling positions of cysteine residues are indicated as C29, C40, C54, C75, C93, and C110. **b** The structure of p27 in complex with Cdk2/cyclin A based on pdb:1JSU[12] and molecular dynamics simulations[17]. Labeled sites are represented as circles, green only for smFA experiments and green/red circles where used in both smFA and smFRET experiments. The p27 residues that are modified by kinases (Y74, Y88, and T187) are shown in orange, purple, and teal spheres, respectively

which Y88 (pY88-p27-KID) or both Y74 and Y88 (pY74/pY88-p27-KID) are phosphorylated, we found that Y88 phosphorylation restored ~20 % of full Cdk2 activity at saturating concentrations of the inhibitor, as previously observed[6], whereas dual Y phosphorylation restored ~50 % of kinase activity (Fig. 2a, Supplementary Figure 1A). Similar Cdk2 reactivation was observed with Y88- and Y74/Y88-phosphorylated full-length p27 (Supplementary Figure 1A, B).

**p27 is accessible to NRTKs in the presence of Cdk2/cyclin A.** Although both Y88 and Y74 are buried against the surface of Cdk2 in the Cdk2/cyclin A/p27-KID structure (Fig. 1b, PDB 1JSU; 4% and 0% solvent accessible surface area, respectively), we have previously shown that Y88 of p27 was accessible for phosphorylation by ABL kinase in cells and by ABL kinase domain (ABL-KD) in vitro (when bound to Cdk2/cyclin A)[6]. To address the issue of Y74 accessibility, we performed p27-KID phosphorylation assays using Src kinase domain (Src-KD) in the presence of Cdk2/cyclin A. We tested p27-KID, pY88-p27-KID, and the Y88 to phenylalanine mutant (Y88F-p27-KID) and found that both Y74 and Y88 were phosphorylated by Src-KD and that the accessibility of Y74 was largely independent of the phospho-status of Y88 (Fig. 2b). These results indicated that both Y74 and Y88 are accessible for modification by NRTKs in the presence of Cdk2/cyclin A.

**Tyrosine phosphorylation of p27 modulates Cdk2 activity.** Reactivation of Cdk2 through phosphorylation of Y88 stimulates Cdk2-dependent phosphorylation of T187 within the flexible C-terminus of p27 through a pseudo uni-molecular mechanism within the Cdk2/cyclin A/pY88-p27 ternary assembly[6,9]. Because Src-dependent dual Y phosphorylation of p27 is associated with reduced p27 protein levels in breast cancer cell lines[7], and because Cdk2-dependent phosphorylation of T187 is associated with

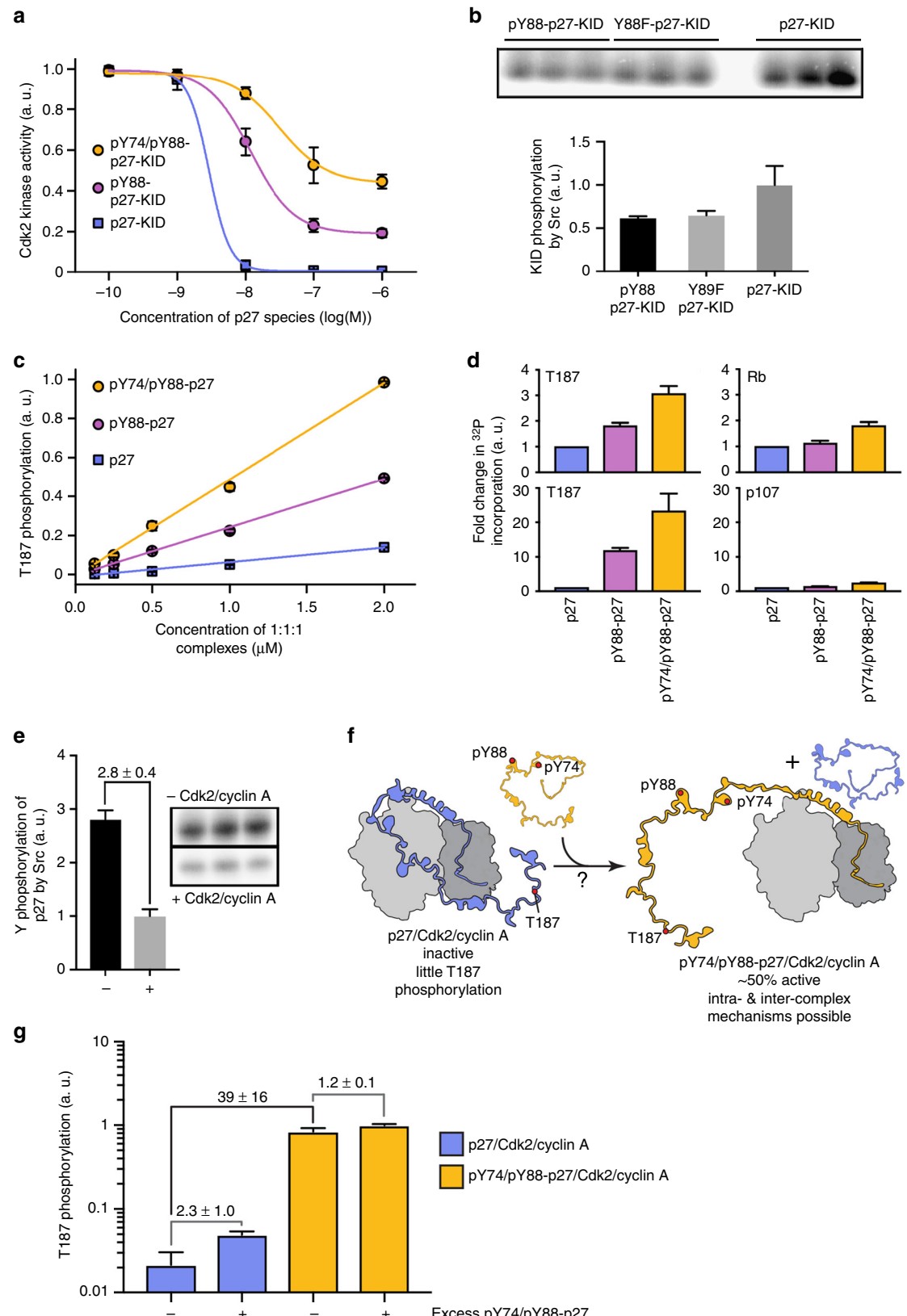

p27 ubiquitination and degradation[10], we next asked if dual Y phosphorylation of p27 within Cdk2/cyclin A complexes would further enhance intra-assembly phosphorylation of T187. To this end, we compared the effect of mono Y and dual Y phosphorylation of p27 on T187 phosphorylation in vitro. As shown previously[6], a low level of T187 phosphorylation can be observed

in 1:1:1 Cdk2/cyclin A/p27 assemblies (Fig. 2c, Supplementary Figure 2A); this arises from bi-molecular reactions (termed the inter-molecular mechanism) between a small amount of free (and active) Cdk2/cyclin A and T187 within either free or Cdk2/cyclin A-bound p27. T187 phosphorylation is enhanced by prior phosphate incorporation into pY88-p27 (Fig. 2c), and dual

**Fig. 2** Incremental Y phosphorylation in p27 exerts rheostat-like control over Cdk2 activity. **a** p27-KID completely inhibits the kinase activity of Cdk2/cyclin A toward histone H1 (squares; IC50 value, $2.8 \pm 0.4$ nM) while pY88-p27-KID (circles; IC50 value, $12 \pm 1$ nM) and pY74/pY88-p27-KID (circles; IC50 value, $29 \pm 7$ nM) are associated with 20% and 43% residual Cdk2 activity at saturating concentrations. **b** Y88 and Y74 are accessible for phosphorylation by Src-KD in the presence of Cdk2/cyclin A. The substrate, pY88-p27-KID, was prepared by prior treatment with ABL-KD. Experiments were performed in triplicate. **c** Phosphorylation of Y88 (purple) and Y74 and Y88 (yellow) enables intra-complex phosphorylation of p27 on T187 within ternary complexes with Cdk2/cyclin A. T187 within unphosphorylated p27 is phosphorylated by Cdk2/cyclin A to a small extent due to inter-molecular reactions, as shown previously[6]. **d** The phosphorylation by Cdk2 of T187 is enhanced to a greater extent by phosphorylation of Y88 or Y74 and Y88 within p27 (that is bound to Cdk2/cyclin A) than phosphorylation of the inter-molecular Cdk2 substrates, Rb and p107. Fold enhancements of phosphorylation of T187 [in the presence of Rb (top) and p107 (bottom)], Rb and p107 are normalized to the extent of phosphorylation of each species observed with unphosphorylated (on Y74 and Y88) p27. For each substrate, the extant of phopshorylation observed with unmodified p27 is set to 1. **e** Phosphorylation of p27 by Src is less efficient in the absence of Cdk2/cyclin A. The gel inset explicitly depicts triplicate experiments in the presence and absence of Cdk2/cyclin A. **f** Scheme depicting the design of the pulse-chase experiment. **g** Pulse-chase experiment showing that the stimulation of *intra*-complex T187 phosphorylation of p27 by tyrosine phosphorylation within the complex is greater than the stimulation of T187 phosphorylation taking place due to the displacement of unmodified p27 by pY74/pY88-p27 from the Cdk2/cyclin A complex. Throughout the figure, error bars represent the standard deviation from the mean (unless otherwise noted, $n = 3$)

Y phosphorylation causes a further ca. Two-fold increase of phosphate incorporation. The data exhibited the linear concentration dependence indicative of an intra-complex mechanism[6] (Fig. 2c). Thus, mono and dual Y phosphorylation of p27 exert rheostat-like control of Cdk2 activity and phosphorylation of p27 on T187.

These observations show that Y phosphorylation of p27 tethered to Cdk2/cyclin A facilitates intra-assembly phosphorylation of T187 (Fig. 2c), whereas results with histone H1 showed facilitation of phosphorylation of an inter-molecular substrate (Fig. 2a). Next, we asked whether reactivation enhances phosphorylation of other physiological substrates of Cdk2, such as retinoblastoma protein (Rb) and p107[11], which are recognized by activated Cdk2/cyclin A via the same hydrophobic surface patch on cyclin A that binds the D1 subdomain of p27-KID. Simultaneous monitoring of phosphorylation of T187 within p27 and of Rb or p107 showed that phosphorylation of Y residues within p27 enhanced Cdk2-dependent phosphorylation of both Rb and p107 (especially upon dual Y74/Y88 phosphorylation of p27), although to a lesser extent than that of the intra-assembly substrate, T187 of p27 (Fig. 2d, Supplementary Figure 2B,C). These results showed that activation of Cdk2 through Y phosphorylation of p27 primarily promoted intra-assembly phosphorylation of T187 but non-co-assembled substrates also gain access to the active site of Cdk2 and/or the substrate-binding pocket on cyclin A.

**Phosphorylation of bound p27 drives downstream signaling**. Although Y74 and Y88 of p27 are significantly buried in complex with Cdk2/cyclin A, they can still be phosphorylated by NRTKs, which would in turn lead to potentiated downstream Cdk2 signaling. To determine if these observations are mechanistically relevant, we compared the extent of p27 tyrosine phosphorylation by Src in the presence and absence of Cdk2/cyclin A and found that under our experimental conditions Src phosphorylation of p27 is ca. 3-fold less efficient in the presence of Cdk2/cyclin A (Fig. 2e). We then performed a pulse-chase experiment in which pY74/pY88-p27 was spiked into preformed ternary complexes of Cdk2/cyclin A with either unmodified p27 or pY74/pY88-p27 along with ATP (Fig. 2f). The excess of uncomplexed pY74/pY88-p27 was used to determine, (i) the efficiency inter-molecular phosphorylation of T187 phosphorylation by the small amount of Cdk2/cyclin A that is in equilibrium with the two forms of p27, (ii) whether pY74/pY88-p27 could exchange with unmodified p27 complexes with Cdk2/cyclin A on the timescale of the experiment (1 h) so as to promote T187 phosphorylation through the intra-complex mechanism, and (iii) whether partially active Cdk/cyclin A

bound to pY74/pY88-p27 could efficiently phosphorylate T187 through the inter-complex mechanism. Although a stoichiometric excess of pY74/pY88-p27 modestly increased the levels of T187 phosphorylation observed for both ternary complexes ($2.3 \pm 1.0$ and $1.2 \pm 0.1$ fold for unmodified p27 and pY74/pY88-p27 ternary complexes, respectively), the increase in T187 phosphorylation is over an order of magnitude stronger ($39 \pm 16$) when comparing preformed complexes prepared with unmodified p27 and pY74/pY88-p27. Taken together, these results suggest that the accessibility of the tyrosine residues of p27 for phosphorylation by NRTKs within the Cdk2/cyclin A complex is mechanistically important in signaling via phosphorylation of T187 and likely evolved to allow for rheostat-like modulation of Cdk2 activity.

**Mechanism of Cdk2 reactivation by Y phosphorylation of p27**. The data discussed above suggest that p27 bound to Cdk2/cyclin A can sense and integrate activation signals from NRTKs with different substrate specificities, such as BCR-Abl and Src, which have specificity for Y88 only or both Y74 and Y88, respectively. However, due to the tight binding of p27 to Cdk2/cyclin A, the structural mechanism that enables these phosphorylation events is unclear. We hypothesized that it could be linked with the distributed interaction interface between p27-KID and Cdk2/cyclin A, which may enable phosphorylation to selectively interfere with small portions of this interface causing transient local (segmental) release of the bound inhibitor.

In accord, 2D NMR analysis previously showed that phosphorylation of Y88 displaces the C-terminal half of the D2 subdomain containing Y88 from the ATP-binding pocket of the kinase[6], whereas dual Y74/Y88 phosphorylation displaces the entire D2 subdomain (Fig. 3a). These results indicate that Y phosphorylation activates Cdk2 by making the active site sterically accessible. Next, we showed that the biochemical effect of this phosphorylation-dependent displacement could be mimicked by deletion of residues 80–94 from p27-KID (p27-KID-ΔC) (Supplementary Figure 3A). In order to further illuminate the mechanism of pY88-dependent partial activation of Cdk2, we determined the crystal structure of the p27-KID-ΔC/Cdk2/cyclin A complex (Supplementary Figure 3B, Supplementary Table 1). Despite accessibility of the active site to ATP, only 20% of full kinase activity was regained because p27-KID-ΔC altered the structure of Cdk2 near the active site. In particular, residues 74–79 of p27-KID-ΔC formed an inter-molecular β-strand with β-strand 2 (β2) of Cdk2, displacing the β1 strand that otherwise forms the G-loop that binds the phosphates of ATP (Fig. 3b, Supplementary Figure 3B). This inter-molecular β-strand can also be observed with p27-KID[12], where it reinforces full

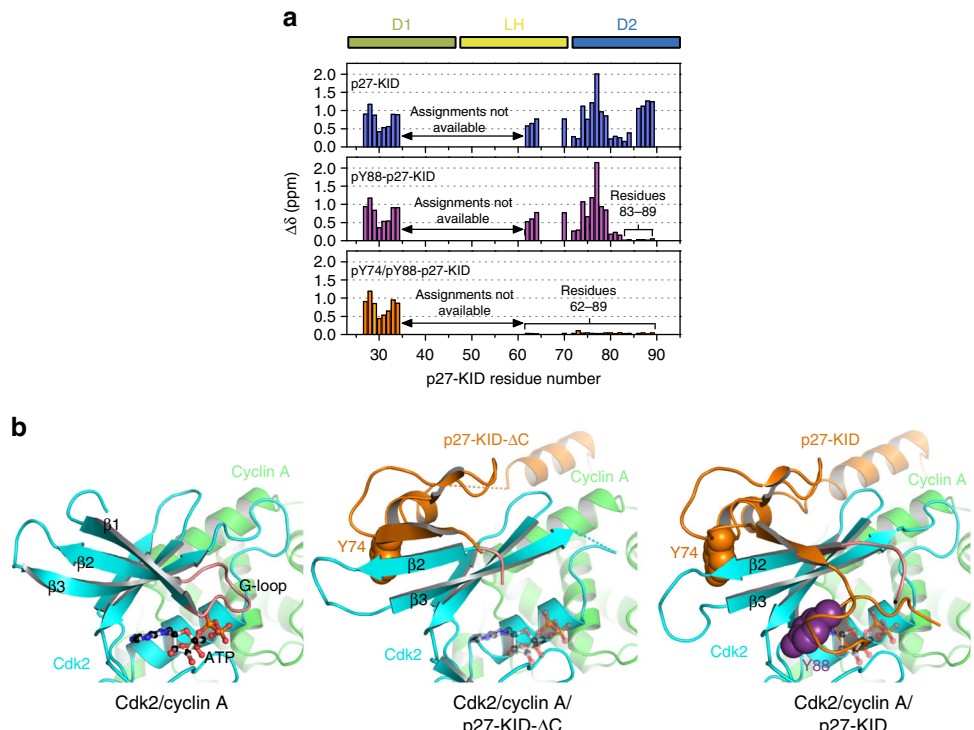

**Fig. 3** Y phosphorylation displaces the D2 region of p27 from Cdk2, mediating reactivation. **a** NMR analysis of the influence of tyrosine phosphorylation on interactions between p27-KID and Cdk2/cyclin A. Chemical shift differences for residues in non-phosphorylated (top) and tyrosine-phosphorylated (pY88-p27-KID, middle; and pY74/pY88-p27-KID, bottom) p27-KID bound to Cdk2/cyclin A. Chemical shift differences were calculated from the free and Cdk2/cyclin A-bound state for each phospho-species. Residues near Y88 and within the entire D2 subdomain adopt free state-like conformations in pY88-p27-KID and pY74/pY88-p27-KID, respectively. $\Delta\delta$ values were calculated using the equation: $\Delta\delta = [(\Delta\delta\ ^1H_N)^2 + 0.0289 \times (\Delta\delta\ ^{15}N_H)^2]^{1/2}$. **b** Views of the region of Cdk2 (cyan) bound by p27 (orange) in various structures of Cdk2/cyclin A: Cdk2/cyclin A/ATP (PDB 1JST, left; ATP is shown in ball and stick format), Cdk2/cyclin A/p27-KID-ΔC (determined in this study, middle; a model of ATP is shown in semi-transparent ball and stick format), and Cdk2/cyclin A/p27-KID (PDB 1JSU; the position of ATP in the absence of p27 is shown in semi-transparent ball and stick format). Both p27-KID-ΔC and p27-KID displace β-strand 1 (β1) from the N-terminal β-sheet of Cdk2 but, due to deletion of residues displaced by phosphorylation of Y88 in p27-KID-ΔC, the active site is accessible to ATP (modeled based upon the Cdk2/cyclin A/ATP structure). However, displacement of the β1 strand of Cdk2 (including residues of the G-loop) in the Cdk2/cyclin A/p27-KID-ΔC structure gives rise to an incompletely formed ATP-binding pocket, consistent with the limited biochemical activity of Cdk2 within this complex

inhibition mediated by Y88. In the case of pY88-p27-KID and p27-KID-ΔC, in which the ATP-binding pocket of Cdk2 is accessible, this feature explains the observed partial catalytic activity of the enzyme.

**Intrinsic dynamics in bound p27 enable Y phosphorylation**. To address the key question of how Y74 and Y88, bound within the tight p27-Cdk2/cyclin A assembly, become accessible for phosphorylation by Src-KD, we introduced fluorescent dyes at several positions within p27, using native and non-native cysteine residues (cf. Figure 1b), and monitored their local and global structural and dynamic features using single-molecule multiparameter fluorescence detection (smMFD) of freely diffusing protein molecules. To map local flexibility, we labeled single-cysteine residues at positions 29, 40, 54, 75, and 93 of p27 with BODIPY-FL using a short linker (termed p27-C29, p27-C40, p27-C54, p27-C75, and p27-C93, respectively) and studied the distinct constructs by single-molecule fluorescence anisotropy (smFA) experiments. To additionally map global structural dynamics, three p27 constructs were prepared with pairs of Cys residues (at positions 29–54, 54–93, and 75–110) and labelled with Alexa Fluor 488 and Alexa Fluor 647 (termed p27-C29-54, p27-C54-93, and p27-C75-110, respectively) for use in single-molecule Förster resonance energy transfer (smFRET) experiments.

The question of how Y88, buried in the Cdk2/cyclin A complex, becomes accessible for phosphorylation was resolved by analyzing the local flexibility of BODIPY-labeled p27-C93 and pY88-p27-C93 by performing smFA experiments. The two-dimensional histogram of fluorescence anisotropy $r_D$ vs. fluorescence-weighted average lifetime ($\langle\tau_D\rangle_f$) showed a very broad distribution of anisotropy values of single-molecule bursts (Fig. 4a, for all smFA fit parameters, see Supplementary Tables 2-4). We applied probability distribution analysis (PDA) to properly account for the shot noise in the histograms[13]. PDA disclosed two underlying long lived (>1 ms) states—one of high ($r_D^H$, light blue line, more rigid state) and another of low ($r_D^L$, dark blue line, more flexible state) anisotropy. Comparison with more complex models resulted in marginal improvement over the figure of merit $\chi_r^2$; thus, we used the 2-state model throughout this work (Supplementary Table 2). The single-molecule data revealed an equilibrium, where, even in the absence of Y88 phosphorylation, this region samples a minor, flexible state (33%), which increases in population (to 66%) significantly upon Y88 phosphorylation (Fig. 4a). The existence of this lowly populated, solvent-exposed conformer of region 83–89 of p27-KID provides an explanation for the accessibility of Y88 for phosphorylation by BCR-ABL and Src in cells and the corresponding kinase domains in vitro[6,7].

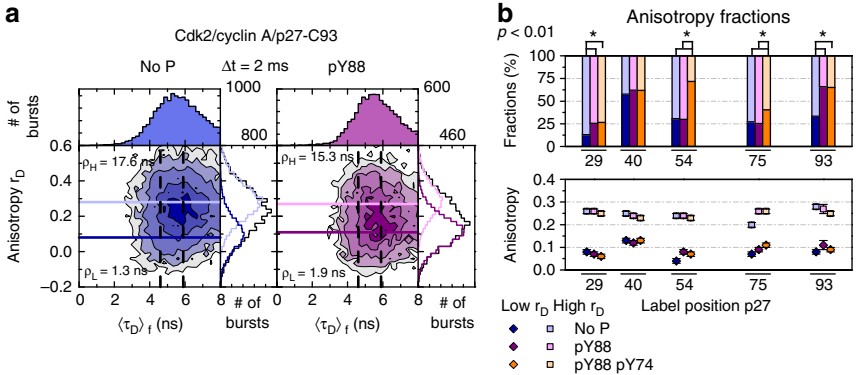

**Fig. 4** Local dynamic fluctuations selected residues within p27 bound to Cdk2/cyclin A detected by fluorescence anisotropy. **a** Time-window analysis ($\Delta t = 2$ ms) for BODIPY-labeled p27-C93 (blue) and pY88-p27-C93 (purple) bound to Cdk2/cyclin A are shown as two-dimensional frequency histograms of scatter-corrected fluorescence anisotropy ($r_D$) vs. the fluorescence-weighted average BODIPY fluorescence lifetime ($\langle \tau_D \rangle_f$). Contours correspond to various levels of occurrences. One-dimensional histograms are the projected distributions over a single variable (right Anisotropy; top $\langle \tau_D \rangle_f$). Number of single-molecule events or # of bursts are shown at the right top corner of each condition. Probability distribution analysis (PDA) is used to fit the $r_D$ distributions with two shot-noise limited states (high $r_D$ and low $r_D$, light and dark colors, respectively). Mean values of high $r_D$ and low $r_D$ are shown as solid lines over the two-dimensional histogram. Vertical dashed lines are identified BODIPY lifetimes. These values are used to calculate the proper rotational correlation times ($\rho_H$ and $\rho_L$ for high $r_D$ and low $r_D$, respectively) for each state using Perrin's equation (Supplementary Equation 9; Supplementary Table 1-3; Supplementary Figure 4). **b** Global fits using PDA for three time-windows ($\Delta t = 1$ ms, 2 ms and 3 ms) are presented for each single Cys variant (C29, C40, C54, C75, and C93). The corresponding fractions and anisotropy values for the low and high $r_D$ are presented in light and dark colors, respectively. The average percentage error over all fractions is 7.0%. Individual errors are found in Supplementary Table 2. $p < 0.01$ significance between fractions are shown with an asterisk

smFA data for p27 labeled with BODIPY on positions C54 or C75 (Fig. 4b) revealed further conformational features of D2 and also of the LH subdomain of p27-KID. In both regions, a major, highly populated state exhibited a high $r_D$ value ($r_D > 0.2$), supporting the view that positions 54 and 75 are mostly rigid both before and after Y88 phosphorylation. The minor state with a low-anisotropy value ($r_D \sim 0.1$; Fig. 4b, Supplementary Figure 4) increased in population upon dual Y74/Y88 phosphorylation of p27, markedly for position 54 and slightly for position 75, consistent with NMR results showing that the entire D2 subdomain was released upon dual Y phosphorylation (Fig. 3a). Similar to the results discussed above for the region containing Y88, the fluctuations of the D2 and LH transiently expose Y74 to solvent and allow for its phosphorylation by NTRKs such as Src. Also, in line with NMR observations are the smFA results for other regions of p27-KID as reported by C40 and C29 within the D1 subdomain. C29, which can be resolved with NMR, exhibits a highly populated rigid (high $r_D$) state, whereas for position C40 (unresolved by NMR) a flexible (low $r_D$) state is the major population. For C40, $r_D$ populations were nearly unchanged upon Y88 and Y88-Y74 phosphorylation, whereas for C29, there is some increase in the flexible state already upon Y88 phosphorylation (Fig. 4b, Supplementary Figure 4).

To elucidate the structural context of the dynamic motions observed by smFA, we performed smFRET experiments and analyzed them by smMFD-plots and correlation analysis (Fig. 5a, Supplementary Figure 5, all determined FRET observables are compiled in Supplementary Tables 5–7). First, we displayed the data by plotting a two-dimensional (2D) smMFD-histogram of burst frequencies for the two FRET indicators, averaged donor-acceptor distance $\langle R_{DA} \rangle_E$ (derived from the intensity ratio of donor over acceptor signal) vs. the average donor fluorescence lifetime $\langle \tau_{D(A)} \rangle_f$, which will be shortened by FRET. This representation allows us to directly assess the structural heterogeneity and dynamics of the sample on different time-scales[14]. The smFRET data for p27-C54-93 bound to Cdk2/cyclin A (Fig. 5a) showed a broadly distributed population peaking at $\langle R_{DA} \rangle_E$ values ~45 Å and $\langle \tau_{D(A)} \rangle_f$ ~2 ns. The maximum of the population in the 2D histogram is shifted to the right of the

static FRET line (Fig. 5a, green curve; Supplementary Table 5), which is a hallmark for fast dynamic mixing of at least two different limiting states[14]. Moreover, the dynamic FRET-populations are very broad, which indicates differently averaged populations in slow exchange. We applied PDA analysis to appropriately account for the broadening of the histograms due to shot noise and distinct acceptor brightness values[13,15]. This way we could reveal at least two averaged conformational states: a highly populated, major state (83%) with $\langle R_{DA} \rangle_E = 45$ Å (Fig. 5a, dark blue), and a somewhat more extended minor state (17%) with $\langle R_{DA} \rangle_E = 52$ Å (Fig. 5a, light blue). We assign the major state to a p27 conformation, in which Y88 is bound within the ATP pocket of Cdk2 [as observed in the Cdk2/cyclin A/p27-KID structure[12]], and the minor state to a conformation in which Y88 was released from this pocket, giving a longer inter-dye distance. This interpretation is confirmed by comparing the measured inter-dye $\langle R_{DA} \rangle_E$ values with expected distances calculated from the X-ray structure of p27-KID bound to Cdk2/cyclin A (1JSU)[12] (Supplementary Table 8), which was extended by molecular dynamics simulations[5] for the unresolved (>90) residues of p27 (Fig. 1b). To describe the dye behavior, we performed coarse grained accessible volume simulations (Supplementary Table 8)[16], and compared the calculated inter-dye distances to the highly populated states of p27-C29-54 (and also of p27-C54-93, and p27-C75-110, see below). In all cases, we found that the smMFD-derived and simulated distances were in agreement (within 10%); hence, this state indeed represents p27 bound to Cdk2/cyclin A.

We next observed that the minor state population increased to 46% upon Y88 phosphorylation. Although ejected due to Y88 phosphorylation based on NMR data (Fig. 3a), 54% of the molecules still exhibited the shorter $\langle R_{DA} \rangle_E$ values (Fig. 5a, b), suggesting that the pY88 region transiently interacts with the ATP pocket. Based on the smFA results, which showed no change after Y88 phosphorylation for position C54 (Fig. 4b), we conclude that the observed structural and local flexibility changes detected by smFRET with pY88-p27-C54-93 occur near Y88, rather than near position 54. Interestingly, despite a significant increase of the low anisotropy state of C54 upon Y74 phosphorylation in smFA (Fig. 4b), smMFD studies with the p27-C54-93 construct showed

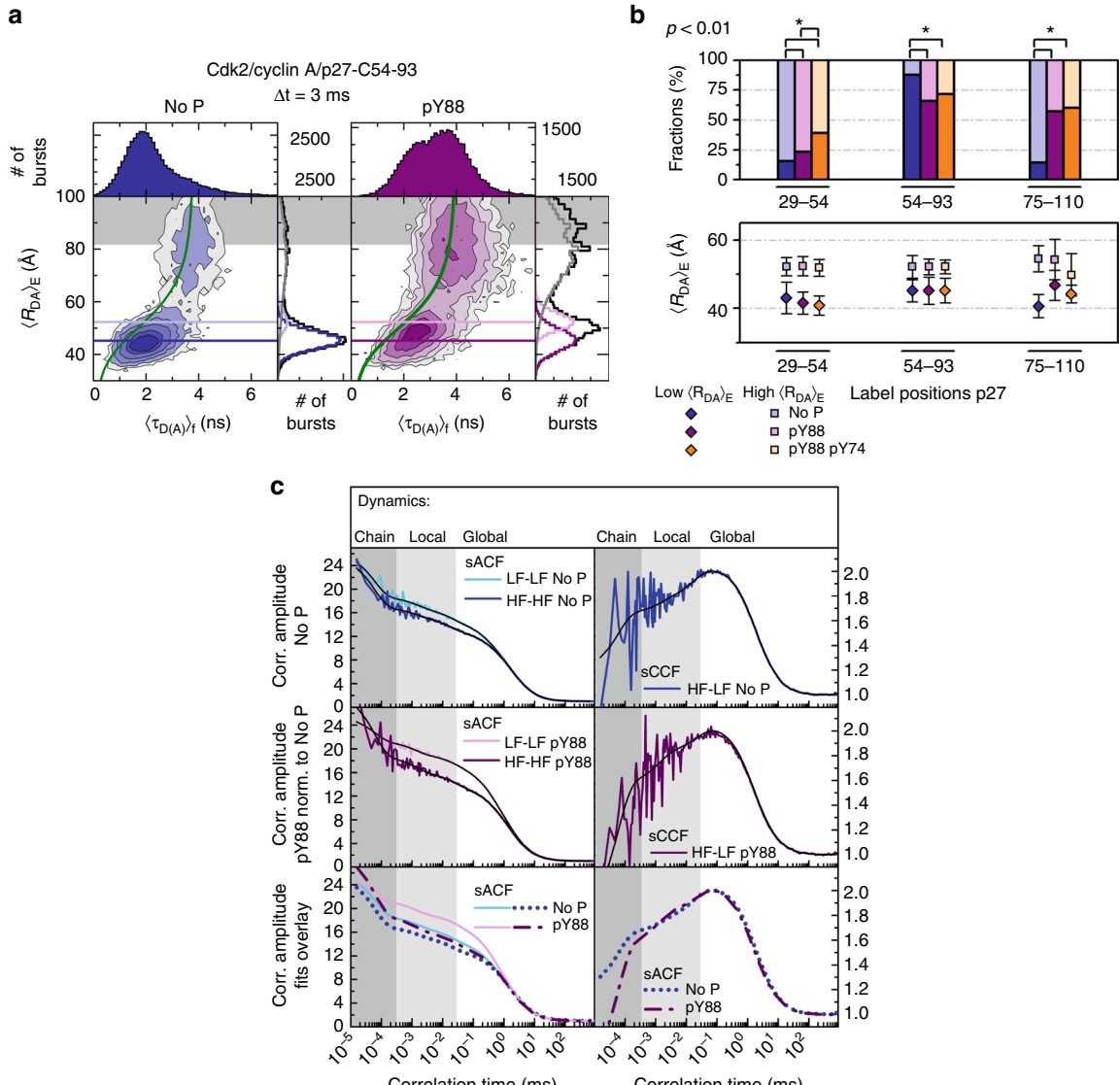

**Fig. 5** Local dynamic fluctuations of selected residues within p27 bound to Cdk2/cyclin A detected by FRET. **a** Time-window analysis ($\Delta t = 3$ ms) for dual-labeled p27-C54-93 (blue) and pY88-p27-C54-93 (purple) bound to Cdk2/cyclin A is shown as two-dimensional frequency histogram of the FRET averaged donor-acceptor distance ($\langle R_{DA} \rangle_E$) vs. the fluorescence-weighted average fluorescence donor lifetime in presence of acceptor ($\langle \tau_{D(A)} \rangle_f$). Gray shaded area indicates area of Donor-only labeled molecules. One-dimensional histograms are the projected distributions over a single variable (right $\langle R_{DA} \rangle_E$, top $\langle \tau_{D(A)} \rangle_f$). PDA is used to fit $\langle R_{DA} \rangle_E$ distributions with two shot-noise limited inter-dye distances (low $\langle R_{DA} \rangle_E$ and high $\langle R_{DA} \rangle_E$, light and dark colors, respectively). Horizontal lines over two-dimensional histogram are added for visual identification of both mean FRET states (Supplementary Table 5-6, Fig. 6). Static FRET lines calculated based on the fluorescence dye properties and PDA are shown in green (Supplementary Table 5): No P: $\langle R_{DA} \rangle_E = (1/(3.8306/(0.0079 * \langle \tau_{D(A)} \rangle_f^2 + 1.0179 * \langle \tau_{D(A)} \rangle_f + -0.1618)-1))^{(1/6)} * 53.0$; pY88: $\langle R_{DA} \rangle_E = (1/(3.9997/(0.0202 * \langle \tau_{D(A)} \rangle_f^2 + 0.9655 * \langle \tau_{D(A)} \rangle_f + -0.1527)-1))^{(1/6)} * 53.0$. Number of bursts are shown at the right top corner of each condition. **b** Global fits using PDA for three time-windows ($\Delta t = 1$ ms, 2 ms, and 3 ms) are presented for each of the FRET-samples (C29-54, C54-93, and C75-110). The corresponding fractions and inter-dye distances for low $\langle R_{DA} \rangle_E$ and high $\langle R_{DA} \rangle_E$ are presented in light and dark colors, respectively. Error bars on $\langle R_{DA} \rangle_E$ represent the half-width of the distribution of each state when fitting with PDA. The average percentage error over all fractions is 1.5%. Individual errors are found in Supplementary Table 6. $p < 0.01$ significance between fractions are shown with an asterisk. **c** The sACFs and sCCFs of filtered fluorescence correlation analysis for p27-C54-93 in complex with Cdk2/cyclin A without phosphorylation and with phosphorylated Y88 map the complex multi-level dynamics of p27 reflecting the change of chain mobility (fit results see Supplementary Table 9)

no major changes in the conformational features or populations of the two states following dual Y74/Y88 phosphorylation in comparison with the mono Y88-phosphorylated form (Fig. 5b; Supplementary Figure 5B). In addition, smFA data for BODIPY-labeled pY74/pY88-p27-C75 indicated only a small change in the populations of the high and low anisotropy states in comparison with the corresponding mono Y-phosphorylated construct (Fig. 4b). In contrast, smFA indicated liberation of position C54

from its more rigid, bound conformation (Fig. 4b). Because NMR results showed the release of the entire D2 subdomain from Cdk2 upon dual Y phosphorylation (Fig. 3a), and smFRET results suggest that the region between C54 and C93 maintains conformational topology similar to that of the bound state, we interpret these results by the existence of a partially populated secondary structure in this region[17] and/or its structural compaction in the free state, which restricts fluctuations of the

residue pair, C54 and C93, and also residue 75. Such compaction driven by hydrophobic residues within D2 subdomain has been observed by NMR[18,19], and is also supported by smFRET data with p27-C75-110. Here, the fraction of molecules with short $\langle R_{DA} \rangle_E$ increased strongly after mono pY88 phosphorylation. As smFA experiments conducted on p27-C75 did not show a change upon Y88 phosphorylation (Fig. 4b), the observed change in distance distribution stems primarily from the liberation of the D2 subdomain, consistent with the NMR data (Fig. 3a).

For comparison, we also studied free, uncomplexed p27 in smFRET, FCS, and smFA experiments. The free protein displayed significant differences in the distributions of the FRET-indicator ($F_D/F_A$), when compared to measurements in complex with Cdk2 and Cyclin A (Supplementary Figure 6). Other controls were also considered (Supplementary Tables 4A and 9). Like in smFA, a marginal improvement in the figure of merit $\chi_r^2$ was not enough justification to add more free parameters on the model function (Supplementary Table 6).

**Bound p27 is intrinsically dynamic across broad timescales**. Irrespective of the actual structural state of liberated D2 subdomain, the coexistence of two anisotropy- and FRET-populations and their positions in the 2D diagrams (Figs. 4a, 5a; Supplementary Figures 4–5) already in the non-phosphorylated complex indicates that p27-KID is in a slow dynamic exchange (on timescales identical to or slower than the diffusion time of ~1.7 ms in the confocal volume, Supplementary Table 10) between a tightly Cdk2/cyclin A-bound state (cf. $K_D$-values in Table 1) and a minor, loosely bound state. To study also the additional fast dynamic processes affecting p27-KID on its transition from the high FRET (HF) to the low FRET (LF) state and within each sub-state, we computed the species-autocorrelation (sACF) and -cross correlation (sCCF) functions of the sub-states depicted in Fig. 4e for the FRET pair p27-C54-C93 (SI Section 1.4)[20]. The recovered dynamic structural fluctuations with relaxation times between 50 ns and 1 ms report on chain (60 ns) and local (~1.5 μs, 20 μs, and 250 μs, for details of the fFCS parameters, cf. Supplementary Table 10) dynamics. The fact that these dynamics are always present, even in the absence of phosphorylation, demonstrates the inherent dynamic anticipation of the complex, which is mani-fested in the liberation of local regions of p27-KID from the Cdk2/cyclin A complex, making Y88 and Y74 accessible for phosphorylation. While for all FRET-pairs the relaxation times change only marginally upon mono or dual phosphorylation (cf. Supplementary Figure 7 and Supplementary Table 10), p27-KID liberation results in a marked increase in the relative fraction of the component showing fast chain dynamics in the sCCF and HF-HF sACF and of slow dynamics in the LF-LF sACF (Fig. 4e, lowest panel). Altogether, smFA and smFRET PDA (Figs. 4a, b and 5a, b) and species correlation functions (Fig. 5c) consistently indicate a gradual shift of the binding mode from the bound state with a higher average FRET efficiency to a more liberated state with a lower average FRET efficiency due to phosphorylation of Y88 and Y74.

In summary, based upon integration of biochemical, NMR and diverse smMFD data, p27 bound to Cdk2/cyclin A is best described as a highly dynamic assembly with multiple conformers in a multi-level energy landscape, whose structural properties and binding interfaces are modulated by the degree of phosphoryla-tion. Prior to phosphorylation, both Y74 and Y88 sample lowly populated solvent-exposed conformations that enable their phosphorylation by NRTKs. Although our results suggest that both Y74 and Y88 can be independently phosphorylated by Src (Fig. 2b), to our knowledge, singly Y74 phosphorylated p27 has not been observed in cells. Additionally, the Src-family kinases that modify Y74 are also known to modify Y88[21]. Therefore, in our experiments we have focused on the pY88 and pY74/pY88-p27 species. Upon Y88 phosphorylation, the region 83–89 is ejected from the ATP-binding pocket of Cdk2, resulting in partial Cdk2 activation[6]. Dual phosphorylation of Y74 and Y88 causes displacement of subdomain D2 from Cdk2 and further kinase activation. Interestingly, the displaced D2 subdomain appears to maintain conformational features similar to the Cdk2-bound state, consistent with past observations with isolated p27-KID[17,18]. A critical feature of the Y phosphorylation-dependent mechanism that modulates p27-dependent regulation of Cdk2 is that both pY88- and pY88/pY74-phosphorylated p27 remain tethered to the Cdk2/cyclin A assembly through binding of the D1 subdomain to cyclin A, enabling intra-assembly phosphoryla-tion of T187 within the flexible p27 C-terminus. These results solve the enigma regarding how Y74 and Y88, which are nearly completely buried in the structure of the p27/Cdk2/cyclin A complex, are made accessible for phosphorylation by NRTKs. These residues dynamically anticipate the Y-phosphorylated state, enabling the critical first step of the multi-step phosphorylation cascade that controls Cdk2 activity and cell division. Further-more, the strength of this phosphorylation signal can be modulated depending on the substrate specificity of the particular NTRK that is activated.

**Thermodynamic and kinetic signatures of the p27 rheostat**. To add further dimensions to this model, we performed surface plasmon resonance (SPR) and isothermal titration calorimetry (ITC) experiments and monitored how Y phosphorylation affects interactions between p27-KID and Cdk2/cyclin A. Phosphoryla-tion of Y residues did not affect the kinetics of p27-KID asso-ciation with Cdk2/cyclin A but did slightly but progressively increase the rate of dissociation (Supplementary Figure 8, Sup-plementary Table 11). These changes were associated with a phosphorylation-dependent increase in apparent $K_D$ values but affinity for Cdk2/cyclin A and cyclin A alone remained high (apparent $K_D$ values of <2.9 nM and <52 nM, respectively), con-sistent with results from NMR and smMFD indicating that the phosphorylated forms of p27 are still bound to the Cdk2/cyclin A complex through the D1 subdomain. Next, ITC was used to determine the number of residues of Y-phosphorylated p27 that fold upon binding to Cdk2/cyclin A, as demonstrated previously for unmodified p27[22]. The extent to which the folding of p27 upon binding is reduced is a surrogate for Y phosphorylation-

**Table 1 Thermodynamic parameters for the binding of p27, Y88E-p27, and Y74E/Y88E-p27 to Cdk2/cyclin A determined by ITC**

| p27 species binding to Cdk2/cyclin A | $K_D$ (nM) | ΔG (kcal mol⁻¹) | ΔH (kcal mol⁻¹) | −TΔS (kcal mol⁻¹) | ΔCp (cal mol⁻¹ K⁻¹) | ℜ (# of residues that fold) |
|---|---|---|---|---|---|---|
| p27 | 4.9 ± 1.5 | −11.4 ± 0.2 | −50.6 ± 1.2 | +39.2 ± 1.2 | −1267 ± 108 | 100 ± 8 |
| Y88E-p27 | 3.4 ± 1.1 | −11.6 ± 0.2 | −38.5 ± 1.1 | +26.9 ± 1.0 | −1119 ± 36 | 81 ± 4 |
| Y74E/Y88E-p27 | 14.0 ± 1.6 | −10.8 ± 0.1 | −34.4 ± 0.5 | +23.6 ± 0.4 | −630 ± 29 | 48 ± 2 |

The ΔCp values were determined from ΔH values measured at 5 °C, 10 °C, 15 °C, 20 °C, and 25 °C and are reported in Supplementary Figure 9.

dependent displacement of p27 from Cdk2/cyclin A. For these experiments, we used constructs in which substitution of Y88 or both Y88 and Y74 with glutamic acids mimicked the biochemical effects of phosphorylation. In agreement with the progressive partial release of p27-KID from Cdk2/cyclin A upon phosphorylation, ITC shows different extents of folding upon binding to Cdk2/cyclin A for p27 and the mono and dual Y phosphomimetic forms, Y88E-p27 and Y74E/Y88E-p27 (Table 1, Supplementary Figure 9). With p27 and Y88E-p27, 99 and 82 residues folded, respectively, upon binding to Cdk2/cyclin A (values of $\mathfrak{R}$, Table 1). This result suggests that a 17 residue-long region of subdomain D2 is displaced from Cdk2 in Y88E-p27, consistent with the biochemical Cdk2 inhibition data, and past SPR and NMR data for p27-KID and pY88-p27-KID[6] on the extent of folding for p27-KID binding to Cdk2/cyclin A[22]. The $K_D$ values for these interactions were identical within error (4.9 ± 1.5 nM and 3.4 ± 1.1 nM, respectively), which suggests a significant decrease in the entropic component and entropy-enthalpy compensation for binding of Y88E-p27 in comparison to p27 ($-T\Delta S$, Table 1). These observations provide strong support for increased dynamics in the bound state upon Y- > E mutation. Consistent with the smFRET experiments showing increased displacement from the Cdk2/cyclin A complex of the mono- and dual Y phosphomimetic forms of p27 (Supplementary Figure 10), the ITC results also show that C-terminal residues outside of the N-terminal KID experienced folding when the KID (comprised of 62 residues) folded upon binding to Cdk2/cyclin A. This is based on the observation that 99 residues of p27 (Table 1), in contrast to 68 residues with p27-KID[22], fold upon binding to Cdk2/cyclin A. An even smaller number of residues folded when dual phosphorylation-mimicking Y74E/Y88E-p27 bound to Cdk2/cyclin A (48 residues; Table 1) and the $K_D$ value increased to 14.0 ± 1.6 nM, consistent with the displacement of the entire D2 region from Cdk2[22]. These conclusions were further supported by NMR data for the two phosphomimetic (Y- > E) p27-KID mutants bound to Cdk2/cyclin A, which showed that the Y88 region and the D2 and LH subdomains, respectively, were displaced from Cdk2 by the Y74E and Y74E/Y88E mutations (Supplementary Figure 11). This displacement of the entire D2 subdomain from Cdk2 upon dual Y phosphorylation of p27 explains substantial reactivation of the kinase (50% of full Cdk2 activity with pY74/pY88-p27-KID and 60% with Y74E/Y88E-p27, Supplementary Figure 11). Thus, the SPR and ITC data, with supporting biochemical and NMR data, support the model discussed above in which Y phosphorylation displaces portions of inhibitory subdomain D2 of p27 from Cdk2, restoring partial kinase activity, while interactions between subdomain D1 and cyclin A are maintained.

**Y phosphorylation of p27 alters interactions with cyclin A.** Our collective results thus far reveal features of the conformational landscape of Cdk2/cyclin A-bound p27 that involve interactions of its D2 subdomain with the kinase subunit of the assembly to regulate kinase activity. However, p27 also inhibits substrate phosphorylation by Cdk2/cyclin A through the binding of its D1 subdomain to a conserved surface on cyclin A, inhibiting substrate recruitment[12,23] (cf. Figure 1). Therefore, we considered the possibility that tyrosine phosphorylation within subdomain D2 could in some way alter interactions between subdomain D1 of p27 and cyclin A and thus affect this second inhibitory mechanism. To test this hypothesis, we utilized p27 constructs labeled on C29 alone (p27–29C) or paired with C54 (p27-C29-54), in smMFD experiments. smFRET experiments showed a major (84%) state with a longer $\langle R_{DA} \rangle_E = 52.3$ Å and a minor (16%) state with a shorter $\langle R_{DA} \rangle_E = 43.1$ Å distance for p27-C29-

54 bound to Cdk2/cyclin A (Fig. 5b, Supplementary Figure 5). After mono and dual phosphorylation, the fraction of the shorter $\langle R_{DA} \rangle_E$ state increased gradually, most probably due to partial release and compaction of the 29–54 region. This is consistent with the dynamic release seen in smFA with BODIPY-labeled p27-C29 bound to Cdk2/cyclin A (Fig. 4b, Supplementary Fig. 4). In all cases, the major state exhibited an $r_D$ value ≥0.25, consistently with C29 binding to cyclin A, whereas the minor state has an $r_D$ value ≤0.08, indicative of increased local mobility relative to the major state. Because BODIPY-labeled p27-C40 remained relatively constant upon mono and dual Y phosphorylation, the changes primarily occur at C29 and the fluctuations of subdomain D1 did not propagate into subdomain LH (Fig. 4b, Supplementary Figure 4B). We suggest that the increased population of the minor state in the smFRET data for the C29-54 dye pair with phosphorylation of both Y74 and Y88 is due to allosteric effects that increase the dynamic fluctuations in the region around position 29. Hence, we term this long-range effect between D1 and D2 cross-complex allostery. This long-range effect may account, in part, for p27 Y phosphorylation-dependent enhanced phosphorylation of the Cdk2 substrates, Rb and p107, and may be consistent with the decreased dependence on the tyrosine phospho-status of p27 (Fig. 2d), as opposed to that of the substrate, histone H1, which lacks an RxL motif that binds to cyclin A[24] (Fig. 2a). Additionally, the increase of Cdk2 phosphorylation of the RXL docked substrates, Rb and p107, may be due to conformational changes propagated through the Cdk2/cyclin A complex that more directly open the hydrophobic patch on cyclin A for substrate docking. The structural results for p27-KID-ΔC bound to Cdk2/cyclin A (Fig. 3b), which mimics the pY88 state as regards Cdk2 activation (Supplementary Fig. 3) and show a cyclin A conformation that is essentially identical to that observed within the p27-KID/Cdk2/cyclin A complex, suggest that such changes would have to involve the C-terminal intrinsically disordered region (IDR) of p27. This IDR is known to dynamically sample many different conformations[5,9] and, through fluctuation proximal to the binding pocket of cyclin A, may influence the binding of the RxL motif within p27 to cyclin A. These hypothetical interactions may change when progressively larger portions of the D2 domain of p27 are displaced from Cdk2 due to tyrosine phosphorylation. However, future studies will be required to test these speculative ideas.

## Discussion

Our study has addressed a key mechanistic question regarding the roles of IDPs as regulatory switches in signaling pathways; specifically, we addressed how sites within a bound IDP (p27 in complex with Cdk2/cyclin A) can become accessible to, and integrate regulatory modifications despite apparent steric inaccessibility within a rigid 3D structure[12]. p27 experiences phosphorylation on two structurally inaccessible tyrosines, which signal p27 degradation and promote cell division[6,7]. This is achieved by the intrinsic dynamics of p27, which, within the p27/Cdk2/cyclin A assembly, is far from being static. By integrating biochemical (enzyme activity and substrate phosphorylation) and biophysical (NMR, X-ray, smMFD, ITC, and SPR) methods, we have discovered that the phosphorylation of tyrosine residues buried within the p27/Cdk2/cyclin A complex is possible because this region structurally fluctuates between a major bound and a minor partially released state compatible with NRTK-mediated phosphorylation (Fig. 6). Upon phosphorylation of either Y88, or Y74 and Y88, the solvent-exposed conformation becomes more populated. Phosphorylation of both residues, which more potently stimulates Cdk2 activity, then displaces the entire D2 subdomain and facilitates intra-complex phosphorylation of

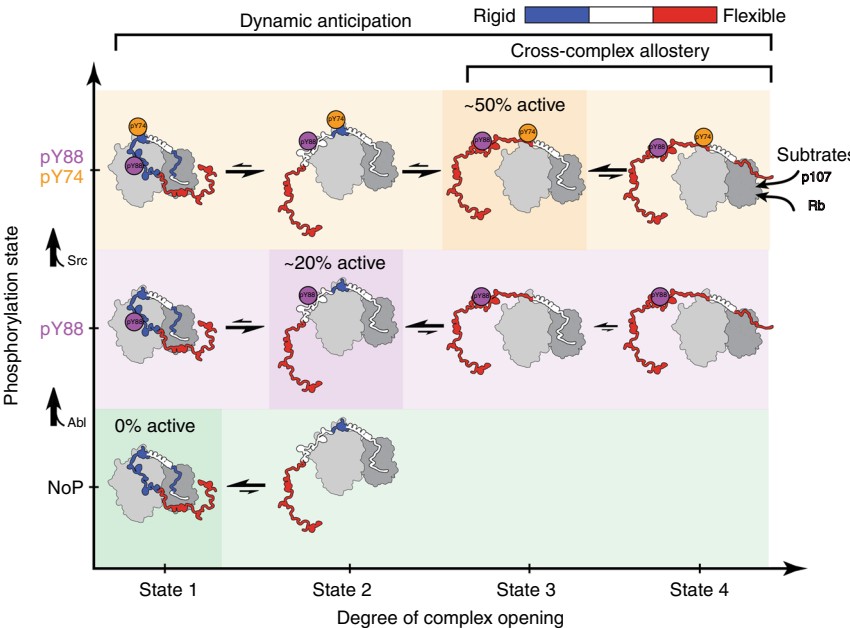

**Fig. 6** Structural dynamics within Cdk2/cyclin A/p27 complex. p27 bound to Cdk2/cyclin A can be best described as a highly dynamic ensemble with multiple conformational states (approximated here by 4) in a multi-level energy landscape, whose structural properties and binding interfaces are tuned by the degree of phosphorylation. Prior to phosphorylation, Y74 and Y88 already sample solvent-exposed conformations that enable its phosphorylation by NRTKs (by Abl). Y88 phosphorylation (marked in purple) partially activates Cdk2. Additional Y74 phosphorylation (by Src, marked in yellow) causes displacement of subdomain D2 from Cdk2 (while preserving its local compact conformation), further kinase activation and intramolecular phosphorylation of T187, also facilitated by partial release of the substrate-binding region on cyclin A. Flexibility is color coded from rigid (blue) to flexible (red)

T187. The activated phosphodegron mediates p27 poly-ubiquitination and degradation and, ultimately, cell-cycle progression. In all, tightly bound p27 within its complex with Cdk2/cyclin A dynamically anticipates the conformational changes caused by Y phosphorylation (Fig. 6).

These structural fluctuations and remodeling of the conformational ensemble enable Cdk2/cyclin A-bound p27 to actively integrate upstream NRTK signals with rheostat-like precision. This rheostat has four settings: with unphosphorylated p27, Cdk2 is off; with pY88-p27, Cdk2 is 20% on; and with pY74/pY88-p27, Cdk2 is 50% on (Fig. 2a), to be turned 100% on by T187 phosphorylation and subsequent elimination of p27 by the ubiquitin-proteasome system. The phosphorylation-dependent functional properties of p27 are reminiscent of observations with the folded protein, dihydrofolate reductase (DHFR), that was shown using NMR spectroscopy to dynamically anticipate a sequence of structural changes that accompanied cofactor binding, substrate binding, and product release[24]. Tuning structural fluctuations by post-translational modification of p27 on Y88 and Y74 through cross-complex allostery may also trigger activation of cyclin A in the Cdk2/cyclin A complex, providing access to its substrate-binding pocket to specific substrates (Fig. 6). This may be reflected in the nonlinear response of substrate modification to p27 phosphorylation, and may be essential for phosphorylation of external substrates, such as Rb and p107.

Post-translational modifications have previously been shown to dramatically alter the conformational landscapes of disordered regions in proteins. For example, phosphorylation of a serine residue within the regulatory light chain of smooth muscle myosin has been shown to stabilize its helical fold[25]. Additionally, serine phosphorylation within a disordered region of the deubiquitinase DUBA modulates the conformational landscape such that a disorder to order transition within its active site occurs only upon binding to ubiquitinated substrates[26]. Strikingly, hyper-phosphorylation of the 4E-BP2 eukaryotic translation initiation factor promotes the transition from a disordered state to a stable

fold, thereby modulating the affinity for its binding partner, eukaryotic translation initiation factor 4E, by over 3 orders of magnitude[27]. In a counter example, phosphomimetic mutation of an interior serine caused the pentamerization domain of the nucleolar protein, Nucleophosmin, to unfold into disordered monomers; this order to disorder transition may be linked to cell-cycle-dependent dissolution of the nucleolus[28,29]. Other examples of roles for post-translational modifications in the regulation of disordered protein regions have been reviewed[30–32]. Extending concepts derived from these many examples, our results demonstrate that solvent inaccessible, unmodified tyrosine residues (Y74 and Y88) in p27 bound to Cdk2/cyclin A transiently fluctuate to become solvent exposed and accessible to non-receptor tyrosine kinases. Phosphorylation of these residues remodels the conformational landscape of bound p27 such that these inhibitory regions are displaced from the kinase, partially activating the kinase. Interestingly, a significant fraction of annotated phosphorylation sites in proteins (ca. 10–20%) are predicted to be inaccessible to solvent[33,34]. These findings, together with our results, suggest that fluctuations to lowly populated, more flexible and accessible states may be a general mechanism for how inaccessible sites of post-translational modification may be modified. Overall, our data are consistent with a model of progressive remodeling of the conformational energy landscape of bound p27 (Fig. 6), and represents another example of how the intrinsic flexibility of IDPs allows them to sense, integrate and propagate information within signaling pathways[35].

## Methods

**Protein expression and purification**. p27 constructs used in this study include residues 22–104 (p27-KID) and 1–198 (full length) of human p27 embedded in a pET28a vector (Novagen)[6]. For all p27 constructs used in this study, tyrosine 89 was mutated to F (Y89F) because cellular ABL phosphorylates only Y88 but the ABL kinase domain (ABL-KD) phosphorylates both Y88 and Y89 in vitro[6]. Non-physiological phosphorylation of Y89 is eliminated with the p27-KID-Y89F construct, hereafter referred to as p27-KID. p27-KID-ΔC was prepared by deletion of residues 80–94 from p27-KID in the pET28a expression vector. To mimic tyrosine

phosphorylation, we prepared constructs with tyrosine 88 (Y88) or both Y88 and tyrosine 74 (Y74) mutated to glutamate (Y88E or Y88E/Y74E) using site-directed mutagenesis. pET28a-based expression vectors for p27 constructs with one or two non-native Cys residues for smMFD experiments were prepared using site-directed mutagenesis starting with p27-Y89F. Primer sequences are shown in Supplementary Table 12. First, four wild-type cysteine (Cys) residues were mutated to Ala or Ser, as follows: Cys 29 to Ala, Cys 49 to Ala, Cys 99 to Ser, and Cys 148 to Ser (termed Cys-less p27). One construct was prepared with only the latter three mutations, leaving a single native Cys residue at position (p27-C29). Then, constructs with one or two non-native Cys residues were prepared in the Cys-less p27 mutant background, as follows: Cys 29 maintained in the Cys-less background, p27-C29; Glu 40 to Cys, p27-C40; Glu 54 to Cys, p27-C54; Glu 75 to Cys, p27-C75; Arg 93 to Cys, p27-C93; Cys 29 and Glu 54 to Cys, p27-C29-54; Glu 54 to Cys and Arg 93 to Cys, p27-C54-93; and Glu 75 to Cys and Ser 110 to Cys, p27-C75-110. The various mono-Cys mutant p27 constructs were shown to exhibit Cdk2/cyclin A inhibitory activity indistinguishable from that of p27-Y89F (Supplementary Figure 15). The various p27 constructs, human Cdk2, T160-phosphorylated Cdk2, and truncated human cyclin A (residues 173–432 of human cyclin A) were expressed in *E. coli* and purified by $Ni^{2+}$-NTA affinity chromatography followed by His-tag removal by incubation with thrombin (Novagen)[22]. p27 constructs were further purified by reverse-phase HPLC over C4 resin. Cdk2 and cyclin A were further purified by gel filtration. The SH3 and kinase domains of murine ABL (termed ABL-KD), and the C-terminal domain of chicken Src (residues 251–533; termed Src-KD) co-expressed with YopH phosphatase in *E. coli* and purified by $Ni^{2+}$-NTA affinity and size exclusion according to established methods[36]. Full-length Src was purchased from ThermoFisher. The C-terminus of human Rb (residues 713–928) was sub-cloned into pET28a with an N-terminal, non-cleavable fusion tag (NusA), expressed and purified from *E. coli* using $Ni^{2+}$-affinity chromatography (termed Rb). The C-terminus of human p107 (residues 385–949) was expressed with glutathione S-transferase (GST) and $(His)_6$ fusion tags in *E. coli* and purified with a GST-affinity column (termed p107). All purification buffers contained 25 mM tris, 100 mM NaCl, and 0.5 mM TCEP at pH 8. Proteins were eluted from $Ni^{2+}$-NTA and GST-affinity resins by including 250 mM imidazole or 10 mM reduced glutathione, respectively. Isotope-labeled samples of p27-KID and p27 for NMR, biochemical, and SPR experiments were prepared in *E. coli* as using MOPS-based minimal media[37] and $^{15}N$-ammonium chloride, $^2H/^{13}C$-glucose, and $^2H_2O$. Phosphorylation of p27 and p27-KID at Y88 was performed by incubating with ABL-KD in the presence of ATP and $Mg^{2+}$ at 37 °C for ca. 1 h. Homogenous preparations of the pY88-p27 and pY88-p27-KID were obtained by purifying the ABL-KD-reacted p27 materials with Phospho-Protein $Fe^{3+}$-affinity resin (Qiagen) according to the manufacturer's instructions. Phosphorylated protein was eluted from $Fe^{3+}$-affinity resin with 50 mM sodium phosphate at pH 7. Doubly phosphorylated, pY74/pY88-p27 was prepared by incubating p27 with repeated additions of Src-KD, ATP, and $Mg^{2+}$ in the presence of 1 mM sodium orthovanadate at 4 °C until the reaction went to completion. Homogeneity of the phosphorylated p27 variants for all biochemical and SPR experiments were confirmed by NMR spectroscopy (Supplementary Figure 12 and 13).

**Cdk2 kinase assays and kinetics of T187 phosphorylation.** The inhibitory profiles of p27 and p27-KID constructs in which either Y88 or Y74 and Y88 were phosphorylated were determined by measuring in vitro Cdk2/cyclin A kinase activity toward histone H1 while titrating the different p27 species in the presence of γ-[$^{32}P$]-ATP[6]. IC50 values were determined using nonlinear regression analysis using Graphpad Prism software. The accessibility of Y74 and Y88 to phosphorylation by Src in the presence of Cdk2/cyclin A was probed by incubating the various KID constructs at 10 μM with 12 μM Cdk2/cyclin A, 36 μM Src-KD, and γ-[$^{32}P$]-ATP for 60 min. Kinetic analysis of the T187 phosphorylation was performed with five different concentrations (0.125, 0.25, 0.5, 1.0, and 2.0 μM) of ternary complexes of p27, pY88-p27, or pY74/pY88-p27 with cyclin A/Cdk2. These reagents were equilibrated with γ-[$^{32}P$]-ATP for four time intervals (20, 40, 60, and 120 min) and were followed by analysis using SDS-PAGE and phosphoimager analysis (GE Healthcare, Piscataway, NJ) of $^{32}PO_4$-T187 within the p27 species. To compare the influence of p27 tyrosine phosphorylation on Cdk2-dependent phosphorylation of the intra-complex substrate, T187, and two intermolecular substrates, Rb and p107, we performed the T187 phosphorylation reactions (using 1.0 and 2.0 μM of the ternary complexes) described above in the presence of equimolar amounts of either Rb or p107 for 40 min in the presence of γ-[$^{32}P$]-ATP. The efficiency of Src phosphorylation of p27 in the presence and absence of Cdk2/cyclin A was assessed by comparing $^{32}P$-incorporation into 25 μM p27 by 1 μM full-length Src and γ-[$^{32}P$]-ATP in the presence and absence of 28 μM Cdk2/cyclin A. To assess the mechanistic relevance of the ability of Y74 and Y88 to be phosphorylated in the presence of Cdk2/cyclin A we prepared ternary (3°) complexes of p27 and pY74/pY88-p27 and assessed the extent of $^{32}P$-incorporation into p27 60 min after addition of γ-[$^{32}P$]-ATP and the presence or absence of 1 excess stoichiometric equivalent of pY74/pY88-p27. The working concentration of the 3° complexes in this experiment was 1 μM. The phosphorylation levels of the various substrates discussed above were determined by following $^{32}P$ incorporation into substrates using SDS-PAGE and phosphoimager analysis (GE Healthcare).

**X-ray crystallography.** p27-KID-ΔC and the Cdk2/cyclin A binary complex were mixed at a 1.1:1.0 mole ratio and concentrated to ~15 mg/ml in 20 mM HEPES, pH 7.5, 5 mM DTT, and 300 mM NaCl by ultrafiltration. The ternary Cdk2/cyclin A/p27-KID-ΔC complex was isolated using gel filtration chromatography (Superdex 200, GE Healthcare, Piscataway, NJ) using the same buffer followed by concentration by ultrafiltration to 15 mg/ml. Crystals of Cdk2/cyclin A/p27-KID-ΔC were grown by the hanging drop vapor diffusion method using the following reservoir solution: 0.1 M HEPES, pH 7.2, 1.4 M lithium sulfate, and 4 mM DTT. The hanging drops contained equal volumes of the reservoir and the protein solution (15 mg/ml in gel filtration buffer) and were equilibrated against the reservoir solution at 18 °C. Crystals were cryoprotected with 24% glycerol and flash frozen in liquid nitrogen. Diffraction data were collected at the Southeast Regional Collaborative Access Team (SER-CAT) 22-BM beamline at the Advanced Photon Source and processed using HKL2000[38]. The structure was determined by molecular replacement using MOLREP[39] with the Cdk2/cyclin A/p27-KID structure (PDB ID 1JSU) as a search model. The structure was refined and optimized using REFMAC[40], PHENIX[41], and COOT[42]. Data collection and refinement statistics are summarized in Supplementary Table 1.

**NMR spectroscopy.** $^2H/^{15}N$-labeled p27-KID constructs (p27-KID, pY88-p27-KID, and pY74/pY88-p27) and their complexes with Cdk2/cyclin A were analyzed using NMR spectroscopy at 800 MHz using methods described previously[6]. Briefly, the samples were dissolved in 20 mM potassium phosphate, pH 6.5, 50 mM arginine, 8% (v/v) $^2H_2O$, 5 mM DTT-$D_6$ and 0.02% (w/v) sodium azide. The isolated p27-KID constructs were analyzed using 2D $^1H$-$^{15}N$ HSQC and their complexes with Cdk2/cyclin A using 2D $^1H$-$^{15}N$ TROSY at 35 °C using a Bruker Avance 800 MHz spectrometer equipped with cryogenically cooled TCI probe. The spectra were interpreted based upon previously established resonance assignments[6]. The NMR experiments with the $^2H/^{15}N$-labeled Y88E and Y74E/Y88E variants of p27-KID were performed similarly and resonances assigned by inspection. Assignment ambiguities were resolved through analysis of 3D HNCA, HN(CO)CA, HNCACB, HN(CO)CACB spectra of the isolated $^{13}C/^{15}N$-labeled Y to E mutant p27-KID constructs.

**Surface plasmon resonance.** Binding studies were performed at 25 °C using a BIACORE 3000 optical biosensor (GE Healthcare). Unphosphorylated and tyrosine-phosphorylated p27-KID constructs were covalently attached to a carboxymethyl dextran-coated gold surface (CM-4 Chip; GE Healthcare). The carboxymethyl groups of dextran were activated with *N*-ethyl-*N′*-(3-dimethylaminopropyl) carbodiimide (EDC) and *N*-hydroxysuccinimide (NHS), and p27-KID constructs were attached at pH 4.0 in 10 mM sodium acetate. Any remaining reactive sites were blocked by reaction with ethanolamine.

The kinetics of association and dissociation were monitored at a flow rate of 50 μl/min. Cdk2/cyclin A, cyclin A, and Cdk2 were prepared in 10 mM Tris (pH 8.0), 300 mM NaCl, 1 mM EGTA, 5 mM DTT, 0.1 mg/mL bovine serum albumin, and 0.005% Tween20. Binding was measured for concentration ranges of 391 pM–25 nM for Cdk2/cyclin A, 6.3–400 nM for cyclin A, and 31 nM–2 μM for Cdk2. To account for injection artifacts, a series of sensorgrams was recorded throughout the experiment after injecting only buffer (blank injections). The chip surface was regenerated with a 30 s injection of 6 M guanidine-HCl. Data reported are the difference in SPR signal between the flow cells containing the p27-KID constructs and a reference cell lacking these constructs. Additional instrumental contributions to the signal were removed by subtraction of the average signal of the blank injections from the reference-subtracted signal[43]. Triplicate injections were made at each concentration, and the data were analyzed globally by simultaneously fitting association and dissociation phases at all concentrations using the program Scrubber2 (Version 2.0b, BioLogic Software). The kinetic rate constants were determined by fitting the data to a 1:1 (Langmuir) interaction model. Equilibrium dissociation constants ($K_D$) were calculated as the quotient $k_d/k_a$.

**Isothermal titration calorimetry.** Thermodynamic parameters for the interaction of p27, Y88E-p27, and Y74E/Y88E-p27 with Cdk2/cyclin A were measured using a MicroCal auto-iTC 200 (Malvern Instruments) isothermal titration calorimeter. Protein samples were exchanged into 20 mM HEPES (pH 7.5), 300 mM NaCl, and 2 mM Tris (2-carboxyethyl) phosphine prior to the experiments. Titrations were performed by first injecting 0.5 μl of 10 μM p27 or p27 Y to E variants into a solution of 1 μM Cdk2/cyclin A binary complex, followed by additional 3.1 μl or 2.5 μl injections. Experiments were carried out from 5 to 25 °C in intervals of 5 °C. Results were analyzed using Origin software (OriginLab). Equilibrium dissociation constant values ($K_D$) and thermodynamic parameters were determined fitting the data to a single-site binding model using a nonlinear least-squares fitting algorithm. The reported values are averages and standard deviations of the mean for three replicates. The values of the heat capacity change (ΔCp) associated with the different p27 species binding to Cdk2/cyclin A were determined as the slope of the temperature dependence of the enthalpy change of binding (ΔH) values and the numbers of residues that folded upon binding (ℜ) were determined using the formalism of Spolar and Record[44], as previously described[22].

**Production of dye-labeled p27 for fluorescence studies**. We prepared p27 constructs in which most or all native cysteine (Cys) residues (at positions 29, 49, 99, and 148) were replaced by serine or alanine, and non-native Cys residues were introduced to allow specific labeling with fluorescent dyes. Before labeling the proteins, all buffers were sterile filtered and degassed. p27 was concentrated to 50–70 µM in buffer A (20 mM Tris-HCl pH 8, 10 mM NaCl) with 10 mM DTT. 2.5 ml of concentrated protein was loaded onto PD10 column and the protein was eluted with freshly degassed 3.5 ml of buffer A without DTT. The eluted p27 was first labeled with the acceptor Alexa Fluor 647 maleimide fluorophore (Invitrogen) at a 1:1 ratio, followed by labeling with the donor Alexa Fluor 488 maleimide fluorophore (Invitrogen) at 1:2 ratio. The dual-labeled p27 was separated from the homo-labeled and unlabeled species using ion-exchange chromatography. p27 Y88E and Y88E/Y74E mutants were prepared in the same way. The presence of energy transfer was analyzed upon excitation at 485 nm of the donor and acceptor dyes at 519 nm and 666 nm, respectively, using a Perkin Elmer LS55 Luminescence Spectrometer. Single-cysteine variants for anisotropy measurements were labeled with 2 × -access of BODIPY-FL or Alexa Fluor 488 (Invitrogen) and purified as above. Labeled p27 was then analyzed by gel-filtration chromatography at 1 µM protein concentration; the protein eluted at a volume expected for the monomer without any evidence of oligomerization or degradation. Labeled p27 was phosphorylated with ABL-KD and Src-KD in the same way as non-labelled protein. To ensure the maximal efficiency of phosphorylation, the kinetics of the reaction were monitored; phosphorylated p27 was separated from non-phosphorylated p27 using Pro-Q® Diamond Phosphoprotein Enrichment Kit (Invitrogen) and the results were analyzed by western-blot using Phospho-Tyrosine Mouse mAb (Bioke). The data showed about 98% phosphorylation of fluorescently-labeled proteins, as detailed in Supplementary Methods and Supplementary Figure 14. The homogeneity of phosphorylated p27 vs. non-phosphorylated p27 was also apparent in single-molecule fluorescence experiments, where the two populations could be separately detected.

**Single-molecule multiparameter fluorescence detection**. smMFD for confocal high-precision Förster Resonance Energy Transfer (hpFRET) studies of single-molecules was done using a 485-nm diode laser (LDH-D-C 485 PicoQuant, Germany, operating at 64 MHz, power at objective 110 µW) exciting freely diffusing labeled molecules that passed through a detection volume of the 60×, 1.2 NA collar (0.17) corrected Olympus objective. The emitted fluorescence signal was collected through the same objective and spatially filtered using a 100 µm pinhole, to define an effective confocal detection volume. We used a new detection and data registration scheme to measure dead time free species cross correlation functions. For that, the signal was divided into parallel and perpendicular components and each of them into two different colors ranges (green and red) through dichroic mirrors. Further, the signal of each component was split with 50/50 beam splitters and additionally filtered by band pass filters, HQ 520/35 and HQ 720/150, for green and red, respectively, in front of the corresponding detector. In total eight photon-detectors were used, four for green (τ-SPAD, PicoQuant, Germany) and four for red channels (APD SPCM-AQR-14, Perkin Elmer, Germany). A time-correlated single-photon counting (TCSPC) module 5 (HydraHarp 400, PicoQuant, Germany) with Time-Tagged Time-Resolved (TTTR) mode and 8 synchronized input channels were used for data registration.

For smMFD (smFA and smFRET) measurements, samples were diluted (buffer used was 20 mM HEPES, pH 7.5, 300 mM NaCl, 40 µM TROLOX) to pM concentration assuring ~1 burst per second. In long measurements, to avoid drying out of the immersion water, an oil immersion liquid with refraction index of water was used (Immersol, Carl Zeiss Inc., Germany). NUNC chambers (Lab-Tek, Thermo Scientific, Germany) were used with 500 µL sample volume. Standard controls consisted of measuring water to determine the instrument response function (IRF), buffer for background subtraction, and the nM concentration of green and red standard dyes (Rh110 and Rh101) in water solutions for calibration of green and red channels, respectively. To calibrate the detection efficiencies we used a mixture solution of dual-labeled DNA oligonucleotides with known distance separation between donor and acceptor dyes.

**Fluorescence analysis of smFA and smFRET experiments**. smFA and smFRET experiments were analyzed using smMFD[14]. We used Probability Distribution Analysis (PDA)[13,15] to determine the anisotropy and distance distributions and their corresponding uncertainties. Details on the analysis are given in Supplementary Methods. Filtered Fluorescence Correlation Spectroscopy (fFCS)[20] was used to identify the species-specific interconversion rates in smFRET experiments. Details are given in Supplementary Methods and Tables. The software used to perform the analysis, written in house, can be downloaded from http://www.mpc.hhu.de/software.html.

## Data Availability
The coordinates and structure factors of the Cdk2/cyclin A/p27-KID-ΔC complex have been deposited in the Protein Data Bank under accession code 6ATH. The source data underlying Fig. 2 are provided as a Source Data file. Other data are available from the corresponding authors upon reasonable request.

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

## Acknowledgements

This work was supported by the Odysseus grant G.0029.12 from Research Foundation Flanders (FWO) for P.T. M.T. was supported by a Marie Sklodowska-Curie postdoctoral fellowship (FP7 Grant 330847 – STARIDP). H.S. acknowledges a starting funds from Clemson University NSF (CAREER MCB- 1749778) and US National Institutes of Health (NIH) grants R01MH08192311 and P20GM121342. C.A.M.S. was supported by the ERC Advanced Grant hybridFRET (671208). R.W.K. was supported by US National Institutes of Health (NIH) grants R01CA082491 and R01GM083159, a US National Cancer Institute Cancer Center Support Grant P30CA21765 (at St. Jude Children's Research Hospital), and ALSAC.

## Author contributions

M.T. prepared samples for smFL experiments, participated in smFL measurements, and analyzed smFL data; H.S. performed smFL experiments, analyzed data, and prepared figures; Y.W. performed biochemical assays and NMR experiments, and prepared associated figures, and prepared samples for SPR experiments; S.F. and K.H. analyzed smFL data and prepared associated figures; A.H.P. performed biochemical assays, prepared figures, and assisted in revisions; M-K.Y. and S.W.W. performed X-ray crystallography experiments and structure determination; B.W. performed and analyzed data from SPR experiments and prepared associated tables and figures; M.B.W. conducted SPR experiments and interpreted SPR data; C-G.P. performed molecular biology and protein biochemistry experiments to prepare protein samples for smFL, SPR, ITC, and X-ray diffraction experiments; S.V. performed analyzed data from ITC experiments and prepared associated tables and figures; L.I. performed NMR experiments and prepared associated figures; P.T. supervised M.T. and performed smFL experimental design, analyzed data, and wrote the paper; C.A.M.S supervised H.S. and performed smFL experimental design, analyzed data, and wrote the paper; and R.W.K. supervised the St. Jude co-authors, conceived the project, performed experimental design, analyzed data, and wrote the paper.

## Additional information

**Competing interests:** The authors declare no competing interests.

