## [Peer Review File · Nature Communications]

Reviewers' comments:

Reviewer #1 (Remarks to the Author):

Tsytlonok et al. investigate the binding of p27 to a Cdk2/CylinA complex. They claim that dynamic fluctuations of bound p27 result in a sequential phosphorylation of two of p27's tyrosines. In addition, a cross-complex allostery through tyrosine phosphorylation of p27 is claimed. These are interesting and novel results and concepts. They might influence thinking in the field.

In the following I concentrate on the fluorescence part of this manuscript. Some more general comments are assembled below ("minor points"). As far as I can judge, the anisotropy, single molecule FRET and correlation spectroscopy experiments are performed very well, but I am missing the following crucial points:

- a) It is essential that binding of p27 to the Cdk2/cyclin A complex is shown under single molecule conditions. I cannot find this crucial control. I would e.g. expect in Fig. 4 to see the data for at least one p27 construct on its own contrasted to the complete complex. Some deviations compared to the NMR data could e.g. be explained by some unbound p27.
- b) At several points (e.g. the "fractions") errors are missing and significances are missing all over. The latter is crucial for several of the claims.
- c) Fig. 4A and 4C are fit with 2 states. What is the evidence for 2 states. Why is 1 or 3 states not applicable?
- d) Fig. 5: The authors claim that for the pY88/pY74 state 1 is absent. I see this state both in Fig. 4B and 4D.

Minor points:

1. I do not understand why "dynamic anticipation", "dynamic structural anticipation" and "dynamic fluctuations" are used for the same thing (at least this is how I interpret this). In my view the last term is most clear.
2. I do not understand what the authors want to claim with "rheostat-like control". There are clearly no currents involved and I also do not see any relation to a strong resistance.
3. At several points in the main text it is unclear, if phosphorylated proteins or mimics are used and which mimics are used in the latter case. Indicating this would be very helpful, otherwise the supplement has to be consulted before reading the main text
4. Online Methods: The authors tested the inhibitory activity of their Cys construct, but do not show any data. In my view, this is a very important point and should be shown in the supplement.
5. Fig. 4A: For some numbers it is unclear to which tick on the axis they belong.
6. Fig. 4: Why are sometimes time-windows of 1ms or 2ms or 3ms used?
7. Table 1: What is the error in R (# of residues that fold)?
8. S.1.1. I do not understand the first sentence. When was "burstwise" and when "time-window" selection chosen and why are two selection criteria used? Please do further specify how the 2sigma criteria is applied.
9. S.1.1. (p.4.) I do not fully understand, how the $A_{i,L}$ and $B_{i,L}$ are empirically determined. Could you specify a little more how they are determined from the dye properties? Why are there 4 components for the A's (and C's) and three components for the B's?
10. S.1.1. (p.5) How was the Foerster distance determined and how is $\kappa^2=2/3$ justified as several anisotropies are larger than 0.2?
11. S.1.2. eqn. 11: There is likely a typo in the first exponent.
12. S1.5. eqn. 21 and 22: If I understand this correctly, eqn. 21 gives the geometric mean distance between the AVs of the two dyes, while eqn. 22 takes into account the non-linearity of the efficiency to distance conversion (which results from the experiment). How are these two quantities then related / compared?

Reviewer #2 (Remarks to the Author):

In this manuscript, the authors utilize a combination of biochemical, biophysical, and structural approaches to describe changes in the dynamics of Cdk2/CyclinA-bound p27 upon phosphorylation. The authors conclude that phosphorylation of two tyrosine residues in the KID domain of p27 weakens interactions with Cdk2 through increased flexibility of the D2 region, which further triggers phosphorylation of T187 and subsequent ubiquitination and degradation of p27 to relieve inhibition of Cdk2/Cyclin A. In general, the data presented here are convincing and of interest to a broad audience. However, there are some gaps in the data included in this manuscript that should be addressed to further strengthen the claims made regarding dynamic changes in bound p27.

Additionally, the authors have done a rather poor job of discussing their findings in relation to the large body of work that has been published regarding the role of post-translational modifications in modulating the conformational ensembles of disordered proteins (see Bah, et al. 2016. JBC 291, 6696-6705 for a recent review), as well as well-characterized examples of allostery within IDPs that are quite similar to the supposedly novel "cross-complex allostery" discussed here (a notable example is Ferreon, et al. 2013. Nature 498, 390-394). This results in the perception that the data and corresponding conclusions are being "oversold" to the reader, which detracts from the actual significance of the results.

Specific comments are included below:

- The single molecule fluorescence data presented here are extensive and largely seem to support the model for relief of Cdk2 inhibition proposed by the authors. However, the presentation of these data needs to be improved to be more accessible to the reader. For example, the figure legends for Fig. 4A and 4C (and the corresponding figures in the supplemental data) should be expanded to include all elements of the figure. There are simply too many elements displayed here that are not explained in the manuscript text or the figure legend, and the numeric labels are confusing.

- Additionally, there are some control experiments that should be completed to confirm that mutations required for site-specific dye labeling have not perturbed the native interactions of p27 with Cdk2/cyclin A. How do the cysteine and alanine mutations affect binding affinities? On p. 25, the authors reference "data not shown" for control experiments (kinase assays) involving the mutant proteins. These data need to be included for full transparency.

- While convincing, the "dynamics" information obtained by the analysis of the single molecule fluorescence data would be further bolstered by the inclusion of supporting NMR data. Clearly the authors are able to obtain NMR samples of p27 in its various phosphorylated states—the fluorescence data shown here would be more impactful if the authors could demonstrate corroborating results obtained by NMR. NMR is not only the current gold standard for obtaining dynamic descriptions of macromolecules in solution, but it would also enable the authors to probe the effects of the phosphorylation events on interactions between p27 and Cdk2/cyclin A on a per residue basis.

- How was the phosphorylation state of p27 samples confirmed? What methods were used to confirm purity and homogeneity of singly and doubly-phosphorylated states?

- Is there any biological evidence to suggest that the sequential phosphorylation of Y88 and Y74 is important? Does mutation of Y74 to F have any effect on the kinetics of Y88 phosphorylation?

- Why were phosphorylated p27 constructs used for some of the experiments, but not for all? Also, why is there no direct comparison for the phosphorylated p27 constructs and glutamic acid phospho-mimicking mutants? It is not unusual to observe discrepancies between affinities determined by SPR and ITC, and as such, it is important to confirm that results for all p27

constructs are consistent between the two methods.

- The authors should be careful with this statement on p. 6 – “The data discussed above suggest that p27 bound to Cdk2/cyclin A can sense and integrate activation signals from different NRTKs.” The results presented here could be specific to the kinases chosen for this study (Src and BCL-Abl).

- The authors should elaborate on the NMR results referenced in this statement from p.11 – “These changes were associated with a phosphorylation-dependent increase in apparent KD values but affinity for Cdk2/cyclin A and cyclin A alone remained high (apparent KD values of <2.9 nM and <52 nM, respectively), consistent with results from NMR and smMFD.”

- The presentation and discussion of the structural results on p.6 should be expanded so that the structural data do not seem like an after-thought. It was not immediately obvious as written that the authors solved a crystal structure of the C-terminal deletion construct of p27-KID to obtain the insights discussed in this section of the manuscript.

- Methods for error calculations are conspicuously absent for some of the biochemical experiments. How were errors determined for the kinase activity assays? What are the error bars showing?

- Legend for Figure 1B – need to include description of how phosphorylation sites are depicted in the figure.

- Legend for Figure 3 – are the chemical shift differences calculated by comparing the bound p27 chemical shifts to the chemical shifts of free p27? This is unclear.

Reviewer #3 (Remarks to the Author):

The manuscript by Tsytlonok and colleagues presents a model describing the phosphorylation of CDK-CyclinA bound inhibitor p27 by tyrosine kinases BCR-ABL and Src. These kinases sequentially phosphorylate Y88 and Y74 and promote intra-assembly phosphorylation (of p27) by Cdk2 on distal T187. The authors use a blend of biochemical, biophysical and single-molecule fluorescence methods to map the conformational dynamics of different regions in Cdk2/CyclinA-bound p27. Based on these multiple approaches the authors find that there is an inherent dynamic anticipation of the complex, making Y88 and Y74 accessible for phosphorylation by the non-receptor tyrosine kinases. Secondly, the data provide structural insights explaining sequential phosphorylation order of Y88 and Y74. Finally, the authors use term “cross-complex allostery” to describe a very interesting finding that binding of the region around Y88 into the ATP pocket of Cdk2 allosterically destabilizes the binding of an N-terminal region of p27 next to another substrate binding pocket on cyclin A.

The study is technically very well designed and performed. The data are convincing and considerably contribute to our understanding of cell cycle regulation at G1/S transition. I recommend it for publishing in Nature Communication after a revision.

Major points:

1) The authors have not considered a possibility for an alternative, equilibrium model. Although, it is interesting to solve the enigma regarding how Y88, which directly participates in Cdk2 inhibition, is made accessible for phosphorylation by NRTK, there is also a possibility that majority of the Y phosphorylation takes place at free p27. As p27 fully inhibits the CDK prior to G1/S transition, there must be a considerable fraction of unbound p27 available. Indeed, in Figure 2B, the authors demonstrate that the tyrosine kinase phosphorylates free p27 more efficiently than the Cdk2/Cyclin A bound p27. The authors could make a case for the relative importance of p27 phosphorylation within the complex vs phosphorylation of free p27. By taking into account: 1) the

off rate of the complex presented in Suppl. Table 10 ($5 \times 10^{-4} \text{ s}^{-1}$) which is about 3 % complex dissociation per minute; 2) the time span of p27 degradation of about 1 hour at G1/S (Barr AR, Heldt FS, Zhang T, Bakal C, Novák B. *Cell Syst.* 2016 Jan 27;2(1)); and 3) by analogy of Sic1, the fact that the ubiquitinated forms of CDK inhibitor do not accumulate (Kõivomägi et al *Nature* 2011) - one could possibly draw an equilibrium system with synthesis and degradation rates that drive p27 degradation within 1 hour purely by phosphorylation of free p27 and dissociation of the inhibitory complex as the rate limiting step. The authors should make a strong argument that Y phosphorylation within the complex is really important and not a side-stream.

2) In connection with the previous point, the data in Figure 1B should be properly quantified. The difference in phosphorylation rates of free and CDK-bound p27 should be presented together with the percentage of the substrate consumption to assess the true difference in initial velocity conditions.

3) One of the most interesting outcomes of the study was that tyrosine phosphorylation also allosterically relieves p27-dependent inhibition of substrate binding to Cdk2/cyclin A, a phenomenon the authors term "cross-complex allostery". As the site 110 is close to the RXL motif in p27 shown to bind the hydrophobic patch of Cyclin A, it is highly likely that the allosteric effect releases the RXL of p27, but at the same time opens the hydrophobic patch pocket to other RXL-docked substrates like Rb and p107. The authors mention this possibility: "This long-range effect may account, in part, for p27 Y phosphorylation-dependent enhanced phosphorylation of the Cdk2 substrates, Rb and p107, and may be consistent with the non-linear increase of their phosphorylation upon phosphate incorporation into p27". However, they should also discuss the possibility of direct opening of the docking pocket for the RXL motifs of these substrates. In fact, it would be possible to test this hypothesis by using kinase assays with Rb and p107, and evaluate the effect of RXLs on phosphorylation rates by mutating the motifs in these substrates. If the tyrosine phosphorylated inhibitory complex shows stronger dependence on intact RXLs, it would be clear that the allosteric effect somehow weakens the binding of p27 RXL region and thereby opens the pocket for RXLs from other substrates (without considerably weakening the affinity of the pocket towards generic RXL motifs). Are there any extra interactions upstream of p27 RXL (that are not present in other substrates) that the allostery may disrupt? Such a model would be quite extraordinary.

Minor points:

1) Page 10 : "Phosphorylation of Y residues did not affect the kinetics of p27 association with Cdk2/cyclin A but did slightly but progressively increase the rate of dissociation (Suppl. Figure 7, Suppl. Table 10); similar results were obtained with p27 and p27-KID constructs." In the Suppl. Table 10, there are only values for the KID constructs, while in the Suppl. Figure 7, the panels are labeled as just "p27", but the text indicates that the KID constructs were used. If the results were similar for p27 and p27-KID constructs, the representative graphs and values in the table should be presented side by side.

2) Figure 2C. The Y axis presents rate of phosphorylation in arbitrary units. However, in this context, the phosphorylation stoichiometry of the 1:1:1 complex is important. We cannot estimate here what is the extent of T187 phosphorylation relatively to the complex concentration, and therefore, true comparison of the phosphorylation rates in case of different p27 constructs is not possible.

Overview of revisions:

We apologize that it has taken us almost one year to complete revisions of the manuscript. We were able to address many reviewer comments and concerns through clarification of technical points and minor revision. However, we wanted to address the biochemical comments of Reviewer 3 through new experiments. This required us to regenerate numerous reagents (*e.g.*, Abl and Src kinase domains, mono and dual Y phosphorylated p27, and Cdk2/cyclin A), which turned out to take much longer than initially anticipated due to lab personnel turnover. Dr. Aaron Phillips, a new co-author, led this effort but needed to learn all of the associated biochemical procedures from scratch as the original lead St. Jude author, Dr. Yuefeng Wang, left the lab some years ago. Dr. Phillips's new experiments clarified a few of our previous biochemical results, which has led to a few changes to the associated results, figures and conclusions. However, these changes do not affect the main take-home messages of the paper. We appreciate the patience of the reviewers regarding our revised manuscript. Below, we briefly outline our revision strategy and associated changes to the manuscript.

In the course of performing new biochemical experiments to address comments of Reviewer 3, we found that some technical details regarding tyrosine phosphorylation of p27 required clarification. Specifically, after efforts to reproduce some of the original Src kinase activity data we found that the availability of Y74 for phosphorylation by Src was independent of the phospho-status at Y88. These findings led us to modify the manuscript and explained a few pieces of single-molecule fluorescence data that had previously been puzzling—namely that dyes in close proximity to Y74 report on a minor conformational state whose population remains constant upon phosphorylation at position 88. The most significant revision to the manuscript is that we no longer state that phosphorylation of Y88 and Y74 is sequential. This creates the formal possibility for two mono phosphorylated forms of p27, with either pY88 or pY74. However, mono pY74-p27 has not been observed *in vitro* or in cells. Therefore, we have maintained the focus of the manuscript on only two of the possible phospho-species (pY88 and pY74/pY88), which have been observed *in vitro* and in normal and cancer cells. The changes to the manuscript are noted in the point-by-point responses to reviewer comments and are identified in the revised manuscript.

Additional revisions inspired by Reviewer 3's comments are as follows. As our previous data showed that Src phosphorylation of p27 is much more efficient in the absence of Cdk2/cyclin A, Reviewer 3 was concerned about the mechanistic relevance of the ability of p27's tyrosine residues to be modified by NRTKs while in complex with Cdk2/cyclin A. If *unbound* p27 is modified by NRTKs and then goes on to displace unmodified p27 from Cdk2/cyclin A complexes, then the dynamic fluctuations of the regions of p27 containing the tyrosine residues that we measured using single-molecule fluorescence would simply be a biophysical curiosity rather than mechanistically relevant for Cdk2 signaling in cells. In order to address this concern we designed a pulse-chase type of experiment where we spiked doubly modified pY74/pY88-p27 and ATP into preformed ternary complexes of p27 (both modified and unmodified) bound to Cdk2/cyclin A. The results of this experiment showed that the stimulation of *intra-complex* T187 phosphorylation due to mono or dual tyrosine phosphorylation of p27 (pY88 or pY74/pY88) is much greater than the stimulation associated with the displacement of the

bound, unmodified p27 by dual Y phosphorylated p27. These data are included in new panels in Figure 2 (E, F and G) and are discussed in our point-by-point responses to reviewers, below.

Reviewers' comments:

Reviewer #1 (Remarks to the Author):

Tsytlonok et al. investigate the binding of p27 to a Cdk2/CylinA complex. They claim that dynamic fluctuations of bound p27 result in a sequential phosphorylation of two of p27's tyrosines. In addition, a cross-complex allostery through tyrosine phosphorylation of p27 is claimed. These are interesting and novel results and concepts. They might influence thinking in the field.

Author reply: We thank the reviewer for their thorough analysis of the manuscript and their favorable comments on its potential impact on the field. We also appreciate their careful analysis of the fluorescence experiments and resulting suggestions for improvement.

In the following I concentrate on the fluorescence part of this manuscript. Some more general comments are assembled below ("minor points"). As far as I can judge, the anisotropy, single molecule FRET and correlation spectroscopy experiments are performed very well, but I am missing the following crucial points:

a) It is essential that binding of p27 to the Cdk2/cyclin A complex is shown under single molecule conditions. I cannot find this crucial control. I would e.g. expect in Fig. 4 to see the data for at least one p27 construct on its own contrasted to the complete complex. Some deviations compared to the NMR data could e.g. be explained by some unbound p27.

Author reply: We thank the reviewer for pointing out the missing controls. Although, they were measured, they were not included in the original submitted version. These experimnts included smFRET, FCS, and smFA of free p27 (uncomplexed). See below in Supplemental Figure 6 and selected Supplemental Tables. The following sentence was added on page 10 to accommodate these results [Note: modified text is in purple color]:

“For comparison, we also studied free, uncomplexed p27 in smFRET, FCS, and smFA experiments. The free protein displayed significant differences in the distributions of the FRET-indicator (F_D/F_A), when compared to measurements in complex with Cdk2 and Cyclin A (Suppl. Figure 6).”

In addition, the diffusion time of the BODIPY labeled samples had an average $t_{diff} = 1.18 \pm 0.21$ ms free in solution compared to 1.61 ± 0.09 ms when bound in complex. Similarly, the average anisotropy is almost double when bound in complex.

- i) **smFRET:** There is an obvious difference between free and in complex. Generally speaking, there is an opening that occurs from the free form to the bound complex.

Supplemental Figure 6. Two-dimensional histogram F_D/F_A vs. lifetime of donor in the presence of acceptor $\langle \tau_{D(A)} \rangle_f$ for free and Cdk2/cyclin A/p27 complex with donor and acceptor dyes at various positions. **A)** C29-54, **B)** C54-93 and **C)** C75-110. For all cases one dimensional projections for F_D/F_A and $\langle \tau_{D(A)} \rangle_f$ are also shown. Pure donor and acceptor fluorescence (F_D and F_A) are corrected for background ($\langle B_G \rangle = 1.62$ kHz (free) or 1.57 kHz (bound) **A,B**) or 0.65 kHz for **C**) $\langle B_R \rangle = 1.03$ kHz (free) or 0.94 kHz (bound) **A,B**) or 0.42 kHz **C**), spectral cross-talk ($\alpha = 1.7\%$) and detection efficiency ratio ($g_G/g_R = 0.8$). Static FRET lines are shown in orange.

- ii) **FCS:** Over all samples, there is an increase of the diffusion time when p27 is bound to the complex.

Supplemental Table 9

Samples (No P)	Free t_{diff} [ms]	Cdk2/Cyclin A t_{diff} [ms]
p27 C29	1.08	1.61
p27 C40	1.27	1.63
p27 C54	0.92	1.58
p27 C75	1.14	1.75
p27 C93	1.48	1.50
Average	1.18	1.61
SD	0.21	0.09
Samples (No P)	free	Cdk2/Cyclin A
p27 C54 C29	1.37	1.76
p27 C54 C93	1.35	1.77
p27 C75 C110	1.26	1.52
Average	1.33	1.68
SD	0.06	0.14

- iii) **smFA:** Comparing the average anisotropy ($\langle r_D \rangle = b_1 r_{D,Low} + b_2 r_{D,High}$), there is a significant increase of $\langle r_D \rangle$ when p27 is bound to the complex. $\langle r_D \rangle$ reflects the overall tumbling of the observed molecule. The larger $\langle r_D \rangle$ is the larger the molecule, thus reflecting the larger size in the complex form.

Supplemental Table 4A

Samples (No P)	Free $\langle r_D \rangle$	Cdk2/Cyclin A $\langle r_D \rangle$
p27 C29	0.07	0.24
p27 C40	0.09	0.18
p27 C54	0.09	0.18
p27 C75	0.10	0.16
p27 C93	0.17	0.21
Average	0.10	0.19
stdev	0.04	0.03

b) At several points (e.g. the "fractions") errors are missing and significances are missing all over. The latter is crucial for several of the claims.

Author reply: Errors in fractions were presented in Supplemental Table 2 for BODIPY samples and Supplemental Table 6 for FRET labeled samples. Due to the chosen representation of complementary bar plots in Figures 4B and 4D, these errors cannot be cleanly represented; however, significance $p < 0.01$ were added to Fig 4B,4D as linking lines between conditions marked with an *. We added the following sentences to the captions. Figures were modified to show significances.

Figure 4B. “The average percentage error over all fractions is 7.0 %. Individual errors are found in Supplemental Table 2. $p < 0.01$ significance between fractions are shown with an *”.

Figure 4D. “The average percentage error over all fractions is 1.5 %. Individual errors are found in Supplemental Table 6. $p < 0.01$ significance between fractions are shown with an *”.

c) Fig. 4A and 4C are fit with 2 states. What is the evidence for 2 states. Why is 1 or 3 states not applicable?

Author reply: Our evidence of model comparison is as follows:

- a) Visual inspection of the distribution. If clearly more than a 1-state model is needed, our first model would consider 2-states (e.g. FRET samples).
- b) Visual inspection of weighted residual distribution. If weighted residuals show “structure” depicting the non-randomness of the residual, a more complex model was used.
- c) A good model is considered when the figure of merit $\chi_r^2 \rightarrow 1$.
- d) When two comparative models show $\chi_r^2 \rightarrow 1$, the change in χ_r^2 must significantly decrease χ_r^2 well beyond the 1σ confidence interval over the χ^2 - distribution defined by $\Delta\chi_r^2 = \sqrt{\frac{2}{N}} \approx 0.2$ (Eqn. 15 on SI) (Soong, T.T. *Fundamentals of Probability and Statistics for Engineers* (Wiley, 2004). Here, $N=51$ is the number of fit points or bins for a typical representation. Hence, to satisfy the null hypothesis, we consider appropriate if, $\Delta\chi_r^2 \gg 0.2$.
- e) If $\Delta\chi_r^2$ improved beyond the statistical test, weighted residual should also improve accordingly, and $\chi_r^2 \rightarrow 1$. Otherwise, both models are said to describe considerably well the data, and based on Occam’s razor, we select the model with the lowest number of free parameters.

Based on the listed considerations, and following the comparative Tables R1 and R2, we observe a significant improvement (~10X) of the figure of merit χ_r^2 when using 2-state vs. the 1-state model for the BODIPY samples. When comparing a 2-state vs. a 3-state model, there is only a marginal improvement of χ_r^2 for the BODIPY over the condition e) above. The 2-state model satisfied condition b) and c).

Supplemental Table 2B. Comparative fit statistics between 1, 2, and 3-state model for BODIPY samples.

BODIPY Samples (No P) In complex Cdk2/Cyclin A	χ_r^2				
	1 state	2 states*	3 states	$ \Delta\chi_{r,1-2}^2 $	$ \Delta\chi_{r,3-2}^2 $
p27 C29	17.40	1.26	0.99	16.1	0.27
p27 C40	16.70	1.54	1.24	15.2	0.30
p27 C54	13.50	1.45	1.05	12.1	0.40
p27 C75	8.74	1.26	0.97	7.5	0.29
p27 C93	20.50	1.18	0.91	19.3	0.27
* χ_r^2 as reported on Table S2					

For the FRET samples (Table S6, and R2), χ_r^2 is high for both 2-state and 3-state model, with no major improvement of the distribution of the weighted residuals. In the 3-state model, the additional state had a typical distance larger than 60 Å. It's population contribution was minimal with less than 3% of the overall population. In other words, this minor “improvement” on the model, led us to consider the simplest model of two states as a better representation of FRET samples using PDA.

In addition, one needs to remember that smFRET populations on MFD histograms lie off the FRET static line, meaning that there is indication of fast dynamics occurring at sub millisecond timescales. fFCS corroborated these observations showing exchange process occurring at timescales as fast as 10^{-5} ms. Hence it is to be expected that the 2-state model only serves as a reductionist model that help us describe the polymorphic nature of p27 in complex. As a remark, PDA struggles in capturing fast exchanges processes. Thus, we considered the “simplified 2-state-model of two Gaussian-distributed distances to capture the changes in the conformation of p27 by PDA”. Noting that comparing the agreement of one of the experimentally determined distances to that of the expected interdye distance based on the X-ray structure of the complex (Table S8).

Supplementary Table 6C. Comparative fit statistics between 1, 2, and 3-state model for FRET samples.

FRET Samples (No P) In complex Cdk2/Cyclin A	χ_r^2		
	2 states*	3 states	$ \Delta\chi_{r,3-2}^2 $
p27 C29-54	4.8	4.16	0.64
p27 C54-93	5.3	2.94	2.36
p27 C75-100	5.1	4.19	0.91
* χ_r^2 as reported on Table S6			

Occam's razor comparison of more complex models of Anisotropy and $\langle R_{DA} \rangle_E$ distributions, gave us the confidence to describe the data with a two-state model as the simplest model. Thus, the following sentences were added.

Page 8 Comparison with more complex models resulted in marginal improvement over the figure of merit χ_r^2

Page 10 And like in smFA, a marginal improvement on the figure of merit χ_r^2 was not enough justification to add more free parameters on the fit.

d) Fig. 5: The authors claim that for the pY88/pY74 state 1 is absent. I see this state both in Fig. 4B and 4D.

Author reply: As we tried to keep Fig. 5 simple and clear, we omitted State 1 in the first version of our Fig 5. As requested by the reviewer, State 1 was added to Fig 5. The experimental observations are evidence that upon the increasing phosphorylation state, from none to single and double, the distribution of states is shifted significantly towards a state with higher degree of complex opening. We have also highlighted the major states that contribute to the observed kinase activity.

Figure 5. Structural dynamics within Cdk2/cyclinA/p27 complex. p27 bound to Cdk2/cyclinA can be best described as a highly dynamic ensemble with multiple conformational states (approximated here by 4) in a multi-level energy landscape, whose structural properties and binding interfaces are tuned by the degree of

phosphorylation. Prior to phosphorylation, Y74 and Y88 already sample solvent exposed conformations that enable its phosphorylation by NRTKs (by Abl). Y88 phosphorylation (marked in purple) partially activates Cdk2. Additional Y74 phosphorylation (by Src, marked in yellow) causes displacement of subdomain D2 from Cdk2 (while preserving its local compact conformation), further kinase activation and intramolecular phosphorylation of T187, also facilitated by partial release of the substrate-binding region on cyclin A. Flexibility is color coded from rigid (blue) to flexible (red).

Minor points:

1. I do not understand why "dynamic anticipation", "dynamic structural anticipation" and "dynamic fluctuations" are used for the same thing (at least this is how I interpret this). In my view the last term is most clear.

Author reply: We use the term 'anticipation' as our experiments reveal that the fluctuations of the bound tyrosines allow for accessibility by the kinase. Upon phosphorylation the equilibrium shifts further to the open state. Thus, the unphosphorylated state anticipates the phosphorylated state through dynamic fluctuations. For clarity, we now only use the terms "dynamic fluctuations" and "dynamic anticipation" in the manuscript. We revised one sentence in the 2nd paragraph of the Introduction (page 3).

"Herein, we tested this hypothesis by studying p27 bound to Cdk2/cyclin A by NMR spectroscopy, single-molecule multiparameter fluorescence detection (smMFD) and other biophysical techniques, which revealed dynamic fluctuations, which we refer to as "dynamic anticipation", that allow phosphorylation of two of its tyrosine residues, Y88⁶ and Y74⁷ by the NRTKs, Breakpoint cluster region-Abelson murine leukemia viral oncogene homolog 1 (BCR-ABL) and Src, respectively, which in turn lead to potentiated downstream signaling through the Cdk2/cyclin A axis."

2. I do not understand what the authors want to claim with "rheostat-like control". There are clearly no currents involved and I also do not see any relation to a strong resistance.

Author reply: We are using the term 'rheostat' as an analogy that the activity of Cdk2 can be controlled based on which NRTK is activated, leading to four variable Cdk2 activity 'settings' (e.g., bound to p27, pY88-p27, or pY88/pY74-p27, or in the absence of p27). We respectfully feel that the analogy is appropriate.

The definition of 'rheostat' (according to Google) is: "an electrical instrument used to control a current by varying the resistance". In our analogy, "current" is Cdk2 activity and "resistance" is the extent Cdk2 inhibition imposed by the different forms of p27.

3. At several points in the main text it is unclear, if phosphorylated proteins or mimics are used and which mimics are used in the latter case. Indicating this would be very helpful, otherwise the supplement has to be consulted before reading the main text.

Author reply: The only experiments for which only glutamate mimics were used are the ITC experiments. We have edited the manuscript to clarify this. For example, on page 12 of the Results, the text reads:

“For these experiments, we used constructs in which substitution of Y88 or both Y88 and Y74 with glutamic acids mimicked the biochemical effects of phosphorylation. In agreement with the progressive partial release of p27-KID from Cdk2/cyclin A upon phosphorylation, ITC shows different extents of folding upon binding to Cdk2/cyclin A for p27 and the mono and dual Y phosphomimetic forms, Y88E-p27 and Y74E/Y88E-p27 (Table 1, Suppl. Figure 8). With p27 and Y88E-p27, 99 and 82 residues folded, respectively, upon binding to Cdk2/cyclin A (values of R , Table 1).”

We feel that this language is clear regarding the use of glutamic acid residue to mimic pY in these experiments.

4. Online Methods: The authors tested the inhibitory activity of their Cys construct, but do not show any data. In my view, this is a very important point and should be shown in the supplement.

Author reply: Thank you for this suggestion; we have included data for the the single-Cys mutants showing that the mutations do not alter the Cdk2/cyclin A inhibitory activity of p27. These data are now shown in Suppl. Figure 14.

5. Fig. 4A: For some numbers it is unclear to which tick on the axis they belong.

Author reply: Those ticks correspond to the y-axis, which is labeled “# of Bursts”. For example 800 is for the anisotropy of the No P, while 600 is for the number of Burst on $\langle \tau_D \rangle_f$ for pY88.

6. Fig. 4: Why are sometimes time-windows of 1ms or 2ms or 3ms used?

Author reply: When time-window analysis is used, a global analysis of multiple time-windows are necessary for proper modeling (e.g. Suppl. Table 6). We show representative examples at various time-windows to showcase this need. Although, final displayed selection is arbitrary.

7. Table 1: What is the error in R (# of residues that fold)?

Author reply: We regret omitting these error values in the original version of the manuscript. These were determined from the mean and standard deviation of the mean based upon triplicated determinations of ΔS_{Other} using the Spolar and Record formalism (ref. 35) and are now included in Table 1 on page 23.

8. S.1.1. I do not understand the first sentence. When was "burstwise" and when "time-window" selection chosen and why are two selection criteria used? Please do further specify how the 2sigma criteria is applied.

Author reply: Confocal based single molecule experiments on freely diffusing molecules rely on the identification of individual “bursts” of photons. We called “Burstwise” selection to the select photons emitted from a molecule over the observation time (duration of single molecules). This approach was previously introduced as BIFL or Burst Integrated Fluorescence Lifetime (Eggeling C. et al. PNAS, 95, 1556, 1998). Nonetheless, the number of photons per single molecule varies significantly, due to the stochastic nature of diffusion. Thus, to compare similar events, “time-window” analysis (Kalinin, J. Phys. Chem B, 111, 10253, 2007) was introduced for selecting events with the same duration. Both methods carry the same raw data but serve different purposes. PDA assumes events that are similar, thus time-window analysis is preferred. The 2σ criteria is based on the inter-photon arrival time of a background count-rate. Bursts of photons from single molecule cause a smaller interphoton arrival time than background count rates. Thus, the 2σ selection is an statistical approach to select single molecules from background signal. These steps are characteristic of confocal single-molecule selection that has been established in ~ 20 years of research (Schaffer J. et al., J. Phys Chem, 103, 331 1999). The following words were added on the SI “a count rate that is different by a”.

9. S.1.1. (p.4.) I do not fully understand, how the $A_{i,L}$ and $B_{i,L}$ are empirically determined. Could you specify a little more how they are determined from the dye properties? Why are there 4 components for the A's (and C's) and three components for the B's?

Author reply: The mentioned coefficients are determined based on photophysical parameters of the labeled samples. The procedure on determination of these coefficients was published elsewhere (Sindbert S. et al, JACS, 133, 2463, 2011). This citation was added following the change in text as:

“The determination procedure is found elsewhere”

The exact reasoning on why A's and C's contain a cubic polynomial and B's are quadratic is unclear. These are only numerical approximations, because analytical solutions are not possible to derive. Having said that, we are in the process of providing a more robust and general description with few cases where analytical solutions are possible. However, for purposes of this manuscript the polynomial approximations is enough for generating the FRET lines. As a refreshment, FRET lines are parametric relationships that consider the first and second moment of the fluorescence lifetime distribution in order to describe dynamic averaging effects.

10. S.1.1. (p.5) How was the Foerster distance determined and how is $\kappa^2=2/3$ justified as several anisotropies are larger than 0.2?

Author reply: We appreciate reviewer's concern on the justification of $\kappa^2=2/3$. However, we must note that the reported anisotropies are from Bodipy labeling on single dye experiments. Bodipy is much smaller dye thus enhances anisotropy effects. Contrary, for FRET experiments Alexa dyes are used. Alexa dyes contain much longer linker chemistry; thus,

separating the dye from the surface of the molecule and reducing significantly the fluorescence anisotropy. Benchmark studies (Sindbert S. et al, JACS, 133, 2463, 2011; Kalinin S. Nature Methods, 9, 1218, 2012; Hellenkamp B. et al Nature Methods, 15, 669, 2018) have shown that when using these long linkers dyes, dyes are free to move. Hence, $\kappa^2 \sim 2/3$ is a good approximation with an error of less than 7% in the determined interdye distance (Hellenkamp B. et al Nature Methods, 15, 669, 2018).

11. S.1.2. eqn. 11: There is likely a typo in the first exponent.

Author reply: We appreciate the Reviewer. The typo has been corrected

12. S1.5. eqn. 21 and 22: If I understand this correctly, eqn. 21 gives the geometric mean distance between the AVs of the two dyes, while eqn. 22 takes into account the non-linearity of the efficiency to distance conversion (which results from the experiment). How are these two quantities then related / compared?

Author reply: The reviewer interpretation is correct. Eq. 21 and Eq. 22 can be related if a structure is known or modeled. However multiple distances are required; in that case, an empiric polynomial that relates both can be generated. Care must be taken because this polynomial might not fully report all possible pairs. Nonetheless, the polynomial serves as a guide for directly linking both. Empirically, an average of 2-5 Å difference between these distances has been observed. But larger deviations could occur.

Reviewer #2 (Remarks to the Author):

In this manuscript, the authors utilize a combination of biochemical, biophysical, and structural approaches to describe changes in the dynamics of Cdk2/CyclinA-bound p27 upon phosphorylation. The authors conclude that phosphorylation of two tyrosine residues in the KID domain of p27 weakens interactions with Cdk2 through increased flexibility of the D2 region, which further triggers phosphorylation of T187 and subsequent ubiquitination and degradation of p27 to relieve inhibition of Cdk2/Cyclin A. In general, the data presented here are convincing and of interest to a broad audience. However, there are some gaps in the data included in this manuscript that should be addressed to further strengthen the claims made regarding dynamic changes in bound p27.

Author reply: We thank the reviewer for their careful reading of the manuscript and their generally favorable assessment of its merits. We appreciate the suggestions for its improvement, most of which we have been able to address fully.

Additionally, the authors have done a rather poor job of discussing their findings in relation to the large body of work that has been published regarding the role of post-translational modifications in modulating the conformational ensembles of disordered proteins (see Bah, et al. 2016. JBC 291, 6696-6705 for a recent review), as well as well-characterized examples of allostery within IDPs that are quite similar to the supposedly novel “cross-complex allostery” discussed here (a notable example is Ferreon, et al. 2013. Nature 498, 390-394). This results in the perception that the data and corresponding conclusions are being “oversold” to the reader, which detracts from the actual significance of the results.

Author reply: We have modified our discussion to place these findings in the broader context of how post-translational modifications can modulate the conformational landscape of disordered regions and also draw a connection between our work and proteomic studies that have shown that a significant number of observed phosphorylation sites are not expected to be solvent accessible. The modified text and additional references are shown below (from page 15). Throughout, modifications to the manuscript are shown in purple.

“Post-translational modifications have previously been shown to dramatically alter the conformational landscapes of disordered regions in proteins. For example, phosphorylation of a serine residue within the regulatory light chain of smooth muscle myosin has been shown to stabilize its helical fold.²⁶ Additionally, serine phosphorylation within a disordered region of the deubiquitinase DUBA modulates the conformational landscape such that a disorder to order transition within its active site occurs only upon binding to ubiquitinated substrates.²⁷ Strikingly, hyper-phosphorylation of the 4E-BP2 eukaryotic translation initiation factor promotes the transition from a disordered state to a stable fold, thereby modulating the affinity for its binding partner, eukaryotic translation initiation factor 4E, by over 3 orders of magnitude.²⁸ In a counter example, phospho-mimetic mutation of an interior serine caused the pentamerization domain of the nucleolar protein, Nucleophosmin, to unfold into disordered monomers; this order to disorder transition may be linked to cell cycle-dependent dissolution of the nucleolus.^{29, 30} Other

examples of roles for post-translational modifications in the regulation of disordered protein regions have been reviewed.^{31, 32, 33} Extending concepts derived from these many examples, our results demonstrate that solvent inaccessible, unmodified tyrosine residues (Y74 and Y88) in p27 bound to Cdk2/cyclin A transiently fluctuate to become solvent exposed and accessible to non-receptor tyrosine kinases. Phosphorylation of these residues remodels the conformational landscape of bound p27 such that these inhibitory regions are displaced from the kinase, partially activating the kinase. Interestingly, a significant fraction of annotated phosphorylation sites in proteins (ca. 10 to 20%) are predicted to be inaccessible to solvent.^{34, 35} These findings, together with our results, suggest that fluctuations to lowly populated, more flexible and accessible states may be a general mechanism for how inaccessible sites of post-translational modification may be modified. Overall, our data are consistent with a model of progressive remodeling of the conformational energy landscape of bound p27 (Figure 5), and represents another example of how the intrinsic flexibility of IDPs allows them to sense, integrate and propagate information within signaling pathways.³⁶

Additional References:

26. Espinoza-Fonseca LM, Kast D, Thomas DD. Thermodynamic and structural basis of phosphorylation-induced disorder-to-order transition in the regulatory light chain of smooth muscle myosin. *Journal of the American Chemical Society* 130, 12208-12209 (2008).
27. Huang OW, et al. Phosphorylation-dependent activity of the deubiquitinase DUBA. *Nature structural & molecular biology* 19, 171-175 (2012).
28. Bah A, et al. Folding of an intrinsically disordered protein by phosphorylation as a regulatory switch. *Nature* 519, 106-U240 (2015).
29. Mitrea DM, et al. Self-interaction of NPM1 modulates multiple mechanisms of liquid-liquid phase separation. *Nat Commun* 9, 842 (2018).
30. Mitrea DM, et al. Nucleophosmin integrates within the nucleolus via multi-modal interactions with proteins displaying R-rich linear motifs and rRNA. *eLife* 5, (2016).
31. Mitrea DM, Kriwacki RW. Regulated unfolding of proteins in signaling. *Febs Lett* 587, 1081-1088 (2013).
32. Jakob U, Kriwacki R, Uversky VN. Conditionally and Transiently Disordered Proteins: Awakening Cryptic Disorder To Regulate Protein Function. *Chemical reviews* 114, 6779-6805 (2014).
33. Csizmok V, Forman-Kay JD. Complex regulatory mechanisms mediated by the interplay of multiple post-translational modifications. *Current opinion in structural biology* 48, 58-67 (2018).
34. Sirota FL, Maurer-Stroh S, Eisenhaber B, Eisenhaber F. Single-residue posttranslational modification sites at the N-terminus, C-terminus or in-between: To be or not to be exposed for enzyme access. *Proteomics* 15, 2525-2546 (2015).
35. Vandermarliere E, Martens L. Protein structure as a means to triage proposed PTM sites. *Proteomics* 13, 1028-1035 (2013).
36. Bah A, Forman-Kay JD. Modulation of Intrinsically Disordered Protein Function by Post-translational Modifications. *Journal of Biological Chemistry* 291, 6696-6705 (2016).

Specific comments are included below:

1- The single molecule fluorescence data presented here are extensive and largely seem to support the model for relief of Cdk2 inhibition proposed by the authors. However, the presentation of these data needs to be improved to be more accessible to the reader. For example, the figure legends for Fig. 4A and 4C (and the corresponding figures in the supplemental data) should be expanded to include all elements of the figure. There are simply too many elements displayed here that are not explained in the manuscript text or the figure legend, and the numeric labels are confusing.

Author reply: Thank you for this suggestion; we have added various levels of description on the figure captions.

2- Additionally, there are some control experiments that should be completed to confirm that mutations required for site-specific dye labeling have not perturbed the native interactions of p27 with Cdk2/cyclin A. How do the cysteine and alanine mutations affect binding affinities? On p. 25, the authors reference “data not shown” for control experiments (kinase assays) involving the mutant proteins. These data need to be included for full transparency.

Author reply: Thank you for this suggestion; we have included data for the the single-Cys mutants showing that the mutations do not alter the Cdk2/cyclin A inhibitory activity of p27. These data are now shown in Suppl. Figure 14.

3- While convincing, the “dynamics” information obtained by the analysis of the single molecule fluorescence data would be further bolstered by the inclusion of supporting NMR data. Clearly the authors are able to obtain NMR samples of p27 in its various phosphorylated states—the fluorescence data shown here would be more impactful if the authors could demonstrate corroborating results obtained by NMR. NMR is not only the current gold standard for obtaining dynamic descriptions of macromolecules in solution, but it would also enable the authors to probe the effects of the phosphorylation events on interactions between p27 and Cdk2/cyclin A on a per residue basis.

Author reply: Thank you for this suggestion. In this manuscript we present chemical shift difference data showing how tyrosine phosphorylation leads to ejection of large regions of p27 from the Cdk2/cyclin A complex. We agree that it would be very interesting to analyze the minor, more flexible states of p27 when in complex with Cdk2/cyclin A revealed by the smMFD experiments by NMR; however, we see only a single set of resonances for the bound complex and sample stability and concentration has been limiting in pursuing relaxation experiments that would confirm that there are fluctuations of p27 residues (in the various phospho-forms) within the Cdk2/cyclin A complexes. For example, a significant fraction of p27/Cdk2/cyclin A complexes precipitate from 0.3 mM solutions within 24-36 hours. This makes recording quantitative relaxation data essentially impossible, especially given that preparation of the authentically phosphorylated p27 samples requires repetitive kinase reactions to drive

reactions to completion. It is for these reasons that we turned to smFL methods in the first place.

4- How was the phosphorylation state of p27 samples confirmed? What methods were used to confirm purity and homogeneity of singly and doubly-phosphorylated states?

Author reply: The confirmation of the purity and homogeneity of the phosphorylated samples differ depending on the experiment. The purity and homogeneity of all phosphorylated proteins used in the biochemical and SPR experiments were confirmed by NMR while the purity and homogeneity of the proteins used in smMFD experiments were confirmed by checking that the phosphorylation reactions have reached saturation by western blot with phosphor-specific antibody detection. The SI has been updated to include these data (Suppl. Figures 11-13, pages 38-40 and Suppl. Methods 1.6, page 10).

“S1.6. Stoichiometric phosphorylation of p27 and p27-KID for single-molecule fluorescence studies

For phosphorylating p27 with either Src-KD or SRC-KD plus Abl-KD, the protocol described earlier was used.¹⁵ We have gone at great length to ensure stoichiometric phosphorylation and the absence of non-phosphorylated protein that could interfere with the experiments. First, we have checked the kinetics of p27 FL phosphorylation (reaction carried out as in Suppl. Figure 1 and 2 with p27 FL) and found that it reached saturation and p27 could not be further phosphorylated upon removing kinase and adding fresh enzyme (Suppl. Figure 13A). In the case of p27 KID, phosphorylation by Src-KD causes a shift in the position of the band, and the absence of the protein at the original position shows that there is no non-phosphorylated material left (Suppl. Figure 13B). Second, for several p27 FL fluorescence labelled variants (p27 FL C29, C54 and C93), we have bound phosphorylated protein to a specific phospho-protein binding membrane of the Pro-Q[®] Diamond Phosphoprotein Enrichment Kit (Invitrogen), applied extensive washing to ensure that all non-phosphorylated protein is removed (due to which about 90% of the phosphorylated protein was also lost). As shown by a Western blot stained by Phospho-Tyrosine Mouse mAb (Bioke), the remaining 10% was at least 98% phosphorylated (Suppl. Figure 13C). the remaining amount was still more than enough for the single-molecule fluorescence measurements. The homogeneity of phosphorylated p27 vs. non-phosphorylated was also confirmed by the single-molecule fluorescence experiments, in which the phosphorylated and (very little) non-phosphorylated p27 could be seen separately.”

5- Is there any biological evidence to suggest that the sequential phosphorylation of Y88 and Y74 is important? Does mutation of Y74 to F have any effect on the kinetics of Y88 phosphorylation?

Author reply: Given that Y74 phosphorylation has only been observed in tandem with Y88 phosphorylation (both sites are substrates for Src family kinases) we have only focused on the

pY88, singly phosphorylated state. Additionally, we have not been able to reproduce the initial results that phosphorylation of Y88 stimulates Y74 phosphorylation and therefore have revised the paper extensively. The main results section is shown below (page 4):

Tyrosine residues of p27 are accessible to NRTKs in the presence of Cdk2/cyclin A

Although both Y88 and Y74 are buried against the surface of Cdk2 in the Cdk2/cyclin A/p27-KID structure (Figure 1B, PDB 1JSU; 4 % and 0 % solvent accessible surface area, respectively), we have previously shown that Y88 of p27 was accessible for phosphorylation by ABL kinase in cells and by ABL kinase domain (ABL-KD) *in vitro* (when bound to Cdk2/cyclin A)⁶. To address the issue of Y74 accessibility we performed p27-KID phosphorylation assays using Src kinase domain (Src-KD) in the presence of Cdk2/cyclin A. We tested p27-KID, pY88-p27-KID, and the Y88 to phenylalanine mutant (Y88F-p27-KID) and found that in the presence of Cdk2/cyclin A, both Y74 and Y88 were phosphorylated by Src-KD and that the accessibility of Y74 was largely independent of the phospho-status of Y88 (Figure 2B). These results indicated that both Y74 and Y88 are accessible for modification by NRTKs in the presence of Cdk2/cyclin A.

6- Why were phosphorylated p27 constructs used for some of the experiments, but not for all? Also, why is there no direct comparison for the phosphorylated p27 constructs and glutamic acid phospho-mimicking mutants? It is not unusual to observe discrepancies between affinities determined by SPR and ITC, and as such, it is important to confirm that results for all p27 constructs are consistent between the two methods.

Author reply: The discrepancy here is largely due to the amount of protein required for the triplicate ITC experiments (each performed at multiple temperatures) in comparison with the SPR, smMFD, and biochemical experiments. As the trends in affinities derived from ITC experiments are largely in agreement with results from orthogonal methods, we did not directly test *bona fide* phosphorylated p27 by ITC. As discussed above with regard to the suggested NMR relaxation experiments, preparation of large quantities of authentically Y phosphorylated p27 is impractical. We have shown in Suppl Figure 10 (SI, page 37) that the phosphomimics show similar Cdk2/cyclin A inhibition and partial reactivation as the *bona fide* phosphorylated forms (Figure 2A).

7- The authors should be careful with this statement on p. 6 – “The data discussed above suggest that p27 bound to Cdk2/cyclin A can sense and integrate activation signals from different NRTKs.” The results presented here could be specific to the kinases chosen for this study (Src and BCL-Abl).

Author reply: We have revised this statement to read as follows (page 6).

“The data discussed above suggest that p27 bound to Cdk2/cyclin A can sense and integrate activation signals from NRTKs with different substrate specificities such as BCR-Abl and Src, which have specificity for Y88 only or both Y74 and Y88, respectively.”

8- The authors should elaborate on the NMR results referenced in this statement from p.11 – “These changes were associated with a phosphorylation-dependent increase in apparent K_D values but affinity for Cdk2/cyclin A and cyclin A alone remained high (apparent K_D values of <2.9 nM and <52 nM, respectively), consistent with results from NMR and smMFD.”

Author reply: We revised this statement for accuracy. The affinity remains high, and we know from the NMR and smMFD experiments that p27 remains bound through its D1 subdomain, mainly to cyclin A. The modified text is shown below (page 11):

“To add further dimensions to this model, we performed surface plasmon resonance (SPR) and isothermal titration calorimetry (ITC) experiments and monitored how Y phosphorylation affects interactions between p27-KID and Cdk2/cyclin A. Phosphorylation of Y residues did not affect the kinetics of p27-KID association with Cdk2/cyclin A but did slightly but progressively increase the rate of dissociation (Suppl. Figure 7, Suppl. Table 10). These changes were associated with a phosphorylation-dependent increase in apparent K_D values but affinity for Cdk2/cyclin A and cyclin A alone remained high (apparent K_D values of <2.9 nM and <52 nM, respectively), consistent with results from NMR and smMFD indicating that the phosphorylated forms of p27 are still bound to the Cdk2/cyclin A complex through the D1 subdomain.”

9- The presentation and discussion of the structural results on p.6 should be expanded so that the structural data do not seem like an after-thought. It was not immediately obvious as written that the authors solved a crystal structure of the C-terminal deletion construct of p27-KID to obtain the insights discussed in this section of the manuscript.

Author reply: We thank the reviewer for pointing this out. This section has been revised to explicitly note that we have determined the crystal structure of the p27-KID- Δ C/Cdk2/cyclin A complex. A sentence was added on page 7, as follows:

“Next, we showed that the biochemical effect of this phosphorylation-dependent displacement could be mimicked by deletion of residues 80-94 from p27-KID (p27-KID- Δ C) (Suppl. Figure 3A). In order to further illuminate the mechanism of pY88-dependent partial activation of Cdk2 we determined the crystal structure of the p27-KID- Δ C/Cdk2/cyclin A complex (Suppl. Figure 3B, Suppl. Table 1).”

Author reply: We attempted to crystalize various further truncated p27 constructs as mimics of pY74/pY88-p27 with the ternary complex with Cdk2/cyclin A, but these efforts failed to yield diffraction-quality crystals. Therefore, this section, entitled “Structural mechanism of Cdk2 reactivation by tyrosine phosphorylation of p27”, is somewhat short. However, with the minor revision, we feel it clearly describes the experiments performed and the results obtained, which explain why only 20% Cdk2 activity is regained due to Y88 phosphorylation (or deletion of residues 80-94 in p27-KID- Δ C). We did not want to lengthen the manuscript with a more

extended discussion of the X-ray data that would not add further to our understanding of the tyrosine phosphorylation mechanism.

10- Methods for error calculations are conspicuously absent for some of the biochemical experiments. How were errors determined for the kinase activity assays? What are the error bars showing?

Author reply: All error bars in the biochemical experiments report on the standard deviation from the mean with n=3. The figure legends and methods sections have been updated to reflect this fact.

11- Legend for Figure 1B – need to include description of how phosphorylation sites are depicted in the figure.

Author reply: We regret this omission in the original manuscript. We have updated the legend. Specifically, a new sentence was added, as follows (page 20):

“The p27 residues that are modified by kinases (Y74, Y88, and T187) are shown in orange, purple, and teal spheres, respectively.”

12- Legend for Figure 3 – are the chemical shift differences calculated by comparing the bound p27 chemical shifts to the chemical shifts of free p27? This is unclear.

Author reply: We have clarified the figure legend to note that the chemical shift differences were calculated from free and Cdk2/cyclin A-bound forms of each phospho-species. Specifically, a new sentence was added, as follows (page 21):

“Chemical shift differences were calculated from the free and Cdk2/cyclin A-bound state for each phospho-species.”

Reviewer #3 (Remarks to the Author):

The manuscript by Tsytlonok and colleagues presents a model describing the phosphorylation of CDK-CyclinA bound inhibitor p27 by tyrosine kinases BCR-ABL and Src. These kinases sequentially phosphorylate Y88 and Y74 and promote intra-assembly phosphorylation (of p27) by Cdk2 on distal T187. The authors use a blend of biochemical, biophysical and single-molecule fluorescence methods to map the conformational dynamics of different regions in Cdk2/CyclinA-bound p27. Based on these multiple approaches the authors find that there is an inherent dynamic anticipation of the complex, making Y88 and Y74 accessible for phosphorylation by the non-receptor tyrosine kinases. Secondly, the data provide structural insights explaining sequential phosphorylation order of Y88 and Y74. Finally, the authors use term “cross-complex allostery” to describe a very interesting finding that binding of the region around Y88 into the ATP pocket of Cdk2 allosterically destabilizes the binding of an N-terminal region of p27 next to another substrate binding pocket on cyclin A. The study is technically very well designed and performed. The data are convincing and considerably contribute to our understanding of cell cycle regulation at G1/S transition. I recommend it for publishing in Nature Communication after a revision.

Author reply: We thank the reviewer for their careful reading of the manuscript and their comments on how the results enhance our understanding of cell cycle regulation by p27. We also appreciated their insightful suggestions and comments, which we addressed through new experiments, as discussed below.

Major points:

1) The authors have not considered a possibility for an alternative, equilibrium model. Although, it is interesting to solve the enigma regarding how Y88, which directly participates in Cdk2 inhibition, is made accessible for phosphorylation by NRTK, there is also a possibility that majority of the Y phosphorylation takes place at free p27. As p27 fully inhibits the CDK prior to G1/S transition, there must be a considerable fraction of unbound p27 available. Indeed, in Figure 2B, the authors demonstrate that the tyrosine kinase phosphorylates free p27 more efficiently than the Cdk2/Cyclin A bound p27. The authors could make a case for the relative importance of p27 phosphorylation within the complex vs phosphorylation of free p27. By taking into account: 1) the off rate of the complex presented in Suppl. Table 10 ($5 \times 10^{-4} \text{ s}^{-1}$) which is about 3 % complex dissociation per minute; 2) the time span of p27 degradation of about 1 hour at G1/S (Barr AR, Heldt FS, Zhang T, Bakal C, Novák B. Cell Syst. 2016 Jan 27;2(1)); and 3) by analogy of Sic1, the fact that the ubiquitinated forms of CDK inhibitor do not accumulate (Kõivomägi et al Nature 2011) - one could possibly draw an equilibrium system with synthesis and degradation rates that drive p27 degradation within 1 hour purely by phosphorylation of free p27 and dissociation of the inhibitory complex as the rate limiting step. The authors should make a strong argument that Y phosphorylation within the complex is really important and not a side-stream.

Author reply: We found this suggestion extremely interesting and went to great lengths to convince ourselves that the ability of p27's tyrosine residues to be phosphorylated in the

presence of Cdk2/cyclin A was mechanistically relevant for Cdk2 signaling. We have added several additional experiments to Figure 2 to address this point. We were able to show that there is a ca. 3-fold stimulation in the efficiency of p27 phosphorylation by Src in the absence of Cdk2/cyclin A (versus when p27 is found to Cdk2/cyclin A) (new Figure 2E). We then used a pulse-chase experiment to directly address whether an excess of doubly phosphorylated p27 would displace unmodified p27 from ternary p27/Cdk2/cyclin A complexes to significantly stimulate *intra*-complex T187 phosphorylation over the course of an hour (new figure panels, Figure 2F,G). The addition of one mole equivalent of pY74/pY88-p27 to ternary complexes formed with unmodified p27 resulted in a modest stimulation of the T187 signal (2.3 ± 1.0 fold) while the difference in T187 phosphorylation between ternary complexes prepared with unmodified and pY74/pY88-p27 was (39 ± 16 fold). Taken together, these results indicate that the ability of p27's tyrosine residues to be modified within the complex are likely relevant to the control of Cdk2 signalling rather than a biophysical curiosity.

The results of this experiment are very important but perhaps somewhat nuanced. We provide further analysis below.

1. The small difference between the extent of T187 phosphorylation in the first two "lanes" of Fig. 2G indicates that pY74/pY88-p27 does not displace p27 from Cdk2/cyclin A in the 1 hour period of the experiment; otherwise, the phospho-T187 signal would be much greater due to the much higher efficiency of the intra-complex reaction. The small increase in the phospho-T187 signal that is observed is due to a 2-fold increase in the availability of T187 residues within the flexible C-terminal disordered region of p27; these residues are phosphorylated by the small amount of free Cdk2/cyclin A that is in equilibrium with p27/Cdk2/cyclin A complexes.
2. The small difference between the extent of T187 phosphorylation in the last two "lanes" of Fig. 2G indicates that the intra-complex reaction mechanism dominates. If the inter-complex reaction mechanism contributed significantly, a larger increase in the phospho-T187 signal would be expected. These results suggest strongly that p27 molecules that are Y phosphorylated within a Cdk2/cyclin A complex (by Src or Abl) will experience efficient T187 phosphorylation via the intra-complex mechanism and be ubiquitinated and degraded. Free p27 molecules that are phosphorylated on Y88 or Y88 & Y74 do not "invade" p27/Cdk2/cyclin A complexes quickly enough for T187 phosphorylation to occur to a significant extent.
3. The large difference between the 1st and 3rd "lane: for Fig. 2G further indicates the dominance of the intra-complex mechanism in phosphorylation of T187.

We added a new section under Results that describes these experiments (pages 6-7), as follows (modified text is shown throughout in purple).

"Tyrosine phosphorylation of p27 within Cdk2/cyclin A complexes drives downstream signaling

As discussed above, although Y74 and Y88 of p27 are significantly buried in complex with Cdk2/cyclin A, they can still be phosphorylated by NRTKs which would in turn leading to potentiated downstream Cdk2 signaling. To determine if these observations are mechanistically

relevant we compared the extent of p27 tyrosine phosphorylation by Src in the presence and absence of Cdk2/cyclin A and found that under our experimental conditions Src phosphorylation of p27 is ca. 3-fold less efficient in the presence of Cdk2/cyclin A (Figure 2E). We then performed a pulse-chase experiment in which pY74/pY88-p27 was spiked into preformed ternary complexes (3°) of Cdk2/cyclin A with either unmodified p27 or pY74/pY88-p27 along with ATP (Figure 2F). The excess, uncomplexed pY74/pY88-p27 was used to determine, i) the efficiency inter-complex phosphorylation of T187 phosphorylation by the small amount of Cdk2/cyclin A that is in equilibrium with the two forms of p27, ii) whether pY74/pY88-p27 could exchange with unmodified p27 complexes with Cdk2/cyclin A on the timescale of the experiment (1 hour) so as to promote T187 phosphorylation through the intra-complex mechanism, and iii) whether partially active Cdk2/cyclin A bound to pY74/pY88-p27 could efficiently phosphorylate T187 through the inter-complex mechanism. Although a stoichiometric excess of pY74/pY88-p27 modestly increased the levels of T187 phosphorylation observed for both 3° complexes (2.3 ± 1.0 and 1.2 ± 0.1 fold for unmodified p27 and pY74/pY88-p27 3° complexes, respectively), the increase in T187 phosphorylation is over an order of magnitude stronger (39 ± 16) when comparing preformed complexes prepared with unmodified p27 and pY74/pY88-p27. Taken together, these results suggest that the accessibility of the tyrosine residues of p27 for phosphorylation by NRTKs within the Cdk2/cyclin A complex is mechanistically important in signaling via phosphorylation of T187 and likely evolved to allow for rheostat-like modulation of Cdk2 activity.”

2) In connection with the previous point, the data in Figure 1B should be properly quantified. The difference in phosphorylation rates of free and CDK-bound p27 should be presented together with the percentage of the substrate consumption to assess the true difference in initial velocity conditions.

Author reply: We have been more diligent in defining the y-axes in all of the kinase assay data. None of the kinase assays actually measured the reaction rates. The y-axis of Figure 2C is now labeled as “T187 phosphorylation”, rather than “Rate of T187 phosphorylation”. These experiments were all performed by measuring the amount of ^{32}P -incorporation into substrates separated by SDS-PAGE which only can determine the relative amounts of incorporation of ^{32}P into the substrates. The signal from the free γ - ^{32}P -ATP is much higher than that which is incorporated into substrates, precluding precise quantification of the free γ - ^{32}P -ATP consumed. The image below shows a 2-fold dilution series of the free γ - ^{32}P -ATP signal in a typical reaction after a 100 fold dilution. The free γ - ^{32}P -ATP signal is similar to the scale observed of modified p27 after a 400-fold dilution.

3) One of the most interesting outcomes of the study was that tyrosine phosphorylation also allosterically relieves p27- dependent inhibition of substrate binding to Cdk2/cyclin A, a phenomenon the authors term “cross-complex allostery”. As the site 110 is close to the RXL motif in p27 shown to bind the hydrophobic patch of Cyclin A, it is highly likely that the allosteric effect releases the RXL of p27, but at the same time opens the hydrophobic patch pocket to other RXL-docked substrates like Rb and p107. The authors mention this possibility: “This long-range effect may account, in part, for p27 Y phosphorylation-dependent enhanced phosphorylation of the Cdk2 substrates, Rb and p107, and may be consistent with the non-linear increase of their phosphorylation upon phosphate incorporation into p27”. However, they should also discuss the possibility of direct opening of the docking pocket for the RXL motifs of these substrates. In fact, it would be possible to test this hypothesis by using kinase assays with Rb and p107, and evaluate the effect of RXLs on phosphorylation rates by mutating the motifs in these substrates. If the tyrosine phosphorylated inhibitory complex shows stronger dependence on intact RXLs, it would be clear that the allosteric effect somehow weakens the binding of p27 RXL region and thereby opens the pocket for RXLs from other substrates (without considerably weakening the affinity of the pocket towards generic RXL motifs). Are there any extra interactions upstream of p27 RXL (that are not present in other substrates) that the allostery may disrupt? Such a model would be quite extraordinary.

Author reply: We have expanded the discussion of this point under Results to address the possibility that the reviewer suggests but we feel that further experiments to disentangle the proposed mechanisms are outside the scope of this study. Specifically, the end of the last paragraph under Results now reads as follows (pages 13-14):

“This long-range effect may account, in part, for p27 Y phosphorylation-dependent enhanced phosphorylation of the Cdk2 substrates, Rb and p107, and may be consistent with the non-linear increase of their phosphorylation upon phosphate incorporation into p27 (Figure 2D), as opposed to that of the substrate, histone H1, which lacks an RXL motif that binds to cyclin A²⁴ (Figure 2A). However, the non-linear increase of the RXL docked substrates, Rb and p107, may be due to conformational changes propagated through the Cdk2/cyclin A complex that more directly open the hydrophobic patch on cyclin A for substrate docking. The structural results for p27-KID-ΔC bound to

Cdk2/cyclin A (Fig. 3B), which mimics the pY88 state as regards Cdk2 activation (Suppl. Fig. 3) and show a cyclin A conformation that is essentially identical to that observed within the p27-KID/Cdk2/cyclin A complex, suggest that such changes would have to involve the C-terminal intrinsically disordered region (IDR) of p27. This IDR is known to dynamically sample many different conformations^{5,9} and, through fluctuation proximal to the binding pocket of cyclin A, may influence the binding of the RxL motif within p27 to cyclin A. These hypothetical interactions may change when progressively larger portions of the D2 domain of p27 are displaced from Cdk2 due to tyrosine phosphorylation. However, future studies will be required to test these speculative ideas.”

Minor points:

1) Page 10 : “Phosphorylation of Y residues did not affect the kinetics of p27 association with Cdk2/cyclin A but did slightly but progressively increase the rate of dissociation (Suppl. Figure 7, Suppl. Table 10); similar results were obtained with p27 and p27-KID constructs.” In the Suppl. Table 10, there are only values for the KID constructs, while in the Suppl. Figure 7, the panels are labeled as just “p27”, but the text indicates that the KID constructs were used. If the results were similar for p27 and p27-KID constructs, the representative graphs and values in the table should be presented side by side.

Author reply: We apologize for this error and we thank the reviewer for catching this. All SPR experiments were performed with KID constructs. We have clarified this point in Suppl. Fig. 7 and in the text on page 11, as follows:

“Phosphorylation of Y residues did not affect the kinetics of p27-KID association with Cdk2/cyclin A but did slightly but progressively increase the rate of dissociation (Suppl. Figure 7, Suppl. Table 10).”

2) Figure 2C. The Y axis presents rate of phosphorylation in arbitrary units. However, in this context, the phosphorylation stoichiometry of the 1:1:1 complex is important. We cannot estimate here what is the extent of T187 phosphorylation relatively to the complex concentration, and therefore, true comparison of the phosphorylation rates in case of different p27 constructs is not possible.

Author reply: As discussed above, because these assays are conducted by determining the extent of ³²P incorporation into the relevant substrate it is not feasible to determine the stoichiometric extent of T187 phosphorylation within the complex. The y-axis in this figure has been corrected to ‘ T187 phosphorylation’. The different p27 phosho-forms were analyzed within the same gel, allowing for a relative quantification of the incorporation of ³²P at T187 in p27.

REVIEWERS' COMMENTS:

Reviewer #1 (Remarks to the Author):

All my points are well addressed in the revised version of the manuscript.

In particular, the authors have included several more controls, analyzed the significance, improved the description of their analysis and removed the claim that phosphorylation of Y88 and Y74 is sequential.

Therefore, I support publication of this revised manuscript in Nature Communications.

Reviewer #2 (Remarks to the Author):

This revised manuscript by Tsytlonok, et al. convincingly describes the mechanism by which solvent inaccessible tyrosine residues of p27 are capable of being phosphorylated while bound to Cdk2/CyclinA, thus promoting phosphorylation of T187 and subsequent degradation. The authors utilize a combination of biochemical, biophysical, and structural approaches to describe changes in the dynamics of Cdk2/CyclinA-bound p27 upon phosphorylation and illustrate that intrinsic dynamics of the bound p27 facilitate access of the critical tyrosine residues to the modifying NRTKs. This manuscript is greatly improved from the original submission and will undoubtedly be of interest to a broad audience. The authors more than adequately addressed the concerns of this reviewer from the previous review cycle and I commend their honesty in addressing the difficulties in reproducing and resolving some of the more confusing aspects of their previous submission.

A few minor comments are included below:

- For the T187 phosphorylation data shown in Figure 2C for non-Y phosphorylated p27-K1D, why do the authors conclude that phosphorylation is due to an "inter-molecular" mechanism? Is there any possibility that T187 can be phosphorylated in the absence of tyrosine phosphorylation by an intra-complex mechanism? The authors suggest that the linear concentration dependence is indicative of an intra-complex mechanism later in the paragraph (line 126); to my eye, all of the data shown in Figure 2C are linearly dependent on concentration, and thus it is confusing why the authors assume that non-phosphorylated p27 is modified through a different mechanism.

- Is tyrosine phosphorylation of p27 more or less efficient in the presence of Cdk2/CyclinA? In the text (lines 150-151) you state that Src phosphorylation is less efficient in the presence of Cdk2/CyclinA. This statement agrees with data shown in the blot from Figure 2E but not the graph from the same figure, unless something is labeled incorrectly. What are we really looking at here? This figure in particular could benefit from some additional detail in the legend.

- I believe the mutation referenced in Figure 2B should be labeled Y88F. The labels and the text are inconsistent.

Reviewer #3 (Remarks to the Author):

I appreciate that the authors have addressed several points raised in my review with new experiments and the manuscript is now suitable for publishing in Nature communications.

The study is of high technical quality and will significantly contribute to our understanding of p27 and CDK function at G1/S transition, and will be of interest to a broad range of cell biologists, cancer researchers and structural biologists.

My question about a possible alternative equilibrium mechanism was answered by the pulse-chase type of experiment where doubly modified pY74/pY88-p27 was added into preformed ternary complexes. The results of this experiment clearly showed that the stimulation of intra-complex T187 phosphorylation due to mono or dual tyrosine phosphorylation of p27 is much greater than the stimulation associated with the displacement of the bound, unmodified p27 by dual Y phosphorylated p27.

Some additional points for clarification before publishing:

1) The legend of Figure 2D of the new manuscript version seems to be identical to the initial version, but the figure itself is changed. This change is apparently due to some experimental issues mentioned by the authors in the beginning of the rebuttal letter. However, the authors should explain why the Y axis scale in the new version is so different for incubations with Rb and p107.

Furthermore, In Supplementary Figure 2B,C such huge difference between pT187 and p107 is not visible. Neither there is any indication that the phosphorylation extent of pT187 in two different incubations is that different (upper and lower panel of T187).

2) In the new version (407-410):

„This long-range effect may account, in part, for p27 Y phosphorylation-dependent enhanced phosphorylation of the Cdk2 substrates, Rb and p107, and may be consistent with the non-linear increase of their phosphorylation upon phosphate incorporation into p27 (Figure 2D), as opposed to that of the substrate, histone H1, which lacks an RxL motif that binds to cyclin A (Figure 2A). However, the non-linear increase of the RxL docked substrates, Rb and p107, may be due to conformational changes propagated through the Cdk2/cyclin A complex that more directly open the hydrophobic patch on cyclin A for substrate docking.“

The authors should explain better what they mean under the „non-linear increase of the RxL docked substrates, Rb and p107“. I do not see how Figure 1A shows the lack of non-linear increase in H1 while the Rb and p107 show non-linear increase in Figure 2D. The term „non-linear“ remains very confusing in this context.

3) The Figure 2E legend and the section in the text does not explain the results sufficiently clearly. What are the gel bands in the boxes? What kind of phosphorylation is presented on the left panel? Is this the Y-phosphorylation and T187 summarized (the latter in case of added CDK complex)? There seems to be a bit of a confusing controversy: on the gel panels, the addition of the CDK complex enhances the phosphorylation, but on a bar graph the addition reduces the phosphorylation. There is no explanation, what is the meaning and difference of the bar graph and the gel panel, and if the T187 and Y phosphorylations are summarized in case of the ternary complex. Also, the Figure 2E legend should indicate that the three bands on the gel are triplicates of the same experiment. Possibly, the panel is a simple and clear thing to explain, but the current wording leaves the reader confused.

4) In the new version of the Figure 2G: „(G) The stimulation of Intra-complex T187 phosphorylation of p27 by tyrosine phosphorylation within the complex outcompetes the displacement of unmodified p27 by pY74/pY88-p27 from the Cdk2/cyclin A complex.“ Stimulation of T178 phosphorylation cannot outcompete the displacement. It outcompetes the T187 phosphorylation taking place as a result of the displacement?

Minor points:

1) Figure 2B legend says that the experiments were performed in triplicates. It is better to label these lanes „triplicates“, otherwise it leaves the reader with the impression that the lanes have been left un-labeled by mistake, before the legend is read and guessed that they must be the

triplicates.

2) The following sentence in the Supplementary Figure 2 legend is difficult to understand: "Error bars depict the standard deviation for the replicates at 1.0 and 2.0 μM 1:1:1 complexes and Nus-Rb or p107". Perhaps it should be "...and for incubations containing Nus-Rb or p107"?

REVIEWERS' COMMENTS:

Reviewer #1 (Remarks to the Author):

All my points are well addressed in the revised version of the manuscript. In particular, the authors have included several more controls, analyzed the significance, improved the description of their analysis and removed the claim that phosphorylation of Y88 and Y74 is sequential.

Therefore, I support publication of this revised manuscript in Nature Communications.

Reviewer #2 (Remarks to the Author):

This revised manuscript by Tsytonok, et al. convincingly describes the mechanism by which solvent inaccessible tyrosine residues of p27 are capable of being phosphorylated while bound to Cdk2/CyclinA, thus promoting phosphorylation of T187 and subsequent degradation. The authors utilize a combination of biochemical, biophysical, and structural approaches to describe changes in the dynamics of Cdk2/CyclinA-bound p27 upon phosphorylation and illustrate that intrinsic dynamics of the bound p27 facilitate access of the critical tyrosine residues to the modifying NRTKs. This manuscript is greatly improved from the original submission and will undoubtedly be of interest to a broad audience. The authors more than adequately addressed the concerns of this reviewer from the previous review cycle and I commend their honesty in addressing the difficulties in reproducing and resolving some of the more confusing aspects of their previous submission.

A few minor comments are included below:

- For the T187 phosphorylation data shown in Figure 2C for non-Y phosphorylated p27-KID, why do the authors conclude that phosphorylation is due to an “inter-molecular” mechanism? Is there any possibility that T187 can be phosphorylated in the absence of tyrosine phosphorylation by an intra-complex mechanism? The authors suggest that the linear concentration dependence is indicative of an intra-complex mechanism later in the paragraph (line 126); to my eye, all of the data shown in Figure 2C are linearly dependent on concentration, and thus it is confusing why the authors assume that non-phosphorylated p27 is modified through a different mechanism.

We have shown in <https://doi.org/10.1016/j.cell.2006.11.047>, the 2007 Cell paper, that in the absence of tyrosine phosphorylation of p27, the bimolecular component of the reaction dominates. We have modified the legend to stress this point for clarity: “T187 within unphosphorylated p27 is phosphorylated by Cdk2/cyclin A to a small extent due to intermolecular reactions, as shown previously.⁶”

- Is tyrosine phosphorylation of p27 more or less efficient in the presence of Cdk2/CyclinA? In the text (lines 150-151) you state that Src phosphorylation is less efficient in the presence of Cdk2/CyclinA. This statement agrees with data shown in the blot from Figure 2E but not the graph from the same figure, unless something is labeled incorrectly. What are we really looking at here? This figure in particular could benefit from some additional detail in the legend.

Figures 2E and 2G are different experiments. We distinguish them in the main text. We have modified the y-axis title in 2E to 'Y phosphorylation of p27 by Src' and modified the legend of 2G to stress that these data are from the pulse-chase experiment.

- I believe the mutation referenced in Figure 2B should be labeled Y88F. The labels and the text are inconsistent.

Thanks for noticing this, that is indeed a typo.

Reviewer #3 (Remarks to the Author):

I appreciate that the authors have addressed several points raised in my review with new experiments and the manuscript is now suitable for publishing in Nature communications.

The study is of high technical quality and will significantly contribute to our understanding of p27 and CDK function at G1/S transition, and will be of interest to a broad range of cell biologists, cancer researchers and structural biologists.

My question about a possible alternative equilibrium mechanism was answered by the pulse-chase type of experiment where doubly modified pY74/pY88-p27 was added into preformed ternary complexes. The results of this experiment clearly showed that the stimulation of intra-complex T187 phosphorylation due to mono or dual tyrosine phosphorylation of p27 is much greater than the stimulation associated with the displacement of the bound, unmodified p27 by dual Y phosphorylated p27.

Some additional points for clarification before publishing:

1) The legend of Figure 2D of the new manuscript version seems to be identical to the initial version, but the figure itself is changed. This change is apparently due to some experimental issues mentioned by the authors in the beginning of the rebuttal letter. However, the authors should explain why the Y axis scale in the new version is so different for incubations with Rb and p107.

The Y-axes are different between the Rb and p107 gels because of how the normalization is done. In both gels, the signal from both substrates in the unmodified p27 condition is set to 1. In the p107 gel, since the p27 band is near background, the increase in the p27 signal is much larger than that of the p107 signal. The main point we are arguing is that the pseudo-intramolecular reaction is stimulated by tyrosine phosphorylation of p27 to a greater extent than true intermolecular reactions involving Rb and p107.

We have modified the figure legend in an attempt to make the normalization process more clear. The legend to figure 2D now reads, "The phosphorylation by Cdk2 of T187 is enhanced to a greater extent by phosphorylation of Y88 or Y74 and Y88 within p27 (that is bound to Cdk2/cyclin A) than phosphorylation of the inter-molecular Cdk2 substrates, Rb and p107. Fold enhancements of phosphorylation of T187 [in the presence of Rb (top) and p107 (bottom)], Rb

and p107 are normalized to the extent of phosphorylation of each species observed with unphosphorylated (on Y74 and Y88) p27. For each substrate, the extent of phosphorylation observed with unmodified p27 is set to 1."

Furthermore, In Supplementary Figure 2B,C such huge difference between pT187 and p107 is not visible. Neither there is any indication that the phosphorylation extent of pT187 in two different incubations is that different (upper and lower panel of T187).

We agree, it's really how the data are normalized (with respect to the signal in the unmodified condition) that lead to the discrepancies in the scale. It remains clear, however, that stimulation of the pseudo-intramolecular reaction exceeds that of the in trans reaction between Cdk2 and Rb or p107.

2) In the new version (407-410):

„This long-range effect may account, in part, for p27 Y phosphorylation-dependent enhanced phosphorylation of the Cdk2 substrates, Rb and p107, and may be consistent with the non-linear increase of their phosphorylation upon phosphate incorporation into p27 (Figure 2D), as opposed to that of the substrate, histone H1, which lacks an RxL motif that binds to cyclin A (Figure 2A). However, the non-linear increase of the RXL docked substrates, Rb and p107, may be due to conformational changes propagated through the Cdk2/cyclin A complex that more directly open the hydrophobic patch on cyclin A for substrate docking.“

The authors should explain better what they mean under the „non-linear increase of the RXL docked substrates, Rb and p107“. I do not see how Figure 1A shows the lack of non-linear increase in H1 while the Rb and p107 show non-linear increase in Figure 2D. The term „non-linear“ remains very confusing in this context.

We are arguing that since Rb and p107 contain RxL motifs, perhaps the difference in the stimulation of intra-complex p27 T187 phosphorylation vs inter-molecular phosphorylation of Rb and p107 by Cdk2 upon p27 tyrosine phosphorylation can be explained by the fact that p27 is still bound to the Cdk2/cyclin A complex, preventing complete opening of the cyclin A hydrophobic patch for substrate docking.

We have changed the discussion of these data to the following:

“This long-range effect may account, in part, for p27 Y phosphorylation-dependent enhanced phosphorylation of the Cdk2 substrates, Rb and p107, and may be consistent with the decreased dependence on the tyrosine phospho-status of p27 (Figure 2D), as opposed to that of the substrate, histone H1, which lacks an RxL motif that binds to cyclin A²⁴ (Figure 2A). Additionally, the increase in Cdk2 phosphorylation of the RXL docked substrates, Rb and p107, may be due to conformational changes propagated through the Cdk2/cyclin A complex that more directly open the hydrophobic patch on cyclin A for substrate docking.”

3) The Figure 2E legend and the section in the text does not explain the results sufficiently clearly. What are the gel bands in the boxes? What kind of phosphorylation is presented on the left panel? Is this the Y-phosphorylation and T187 summarized (the latter in case of added CDK complex)? There seems to be a bit of a confusing controversy: on the gel panels, the addition of the CDK complex enhances the phosphorylation, but on a bar graph the addition reduces the

phosphorylation. There is no explanation, what is the meaning and difference of the bar graph and the gel panel, and if the T187 and Y phosphorylations are summarized in case of the ternary complex. Also, the Figure 2E legend should indicate that the three bands on the gel are triplicates of the same experiment. Possibly, the panel is a simple and clear thing to explain, but the current wording leaves the reader confused.

Yes, that is a typo, the gel insets are mislabeled. The data in Figure 2E serves as a control to interpret the separate experiment depicted in 2G. We have changed the y-axis label in 2E to further differentiate it from the data in 2G. The label is now, “Y phosphorylation of p27 by Src”. We apologize for the oversight and thank the reviewer for their careful reading of the manuscript.

4) In the new version of the Figure 2G: „(G) The stimulation of Intra-complex T187 phosphorylation of p27 by tyrosine phosphorylation within the complex outcompetes the displacement of unmodified p27 by pY74/pY88-p27 from the Cdk2/cyclin A complex.“ Stimulation of T178 phosphorylation cannot outcompete the displacement. It outcompetes the T187 phosphorylation taking place as a result of the displacement?

We have changed the legend to Figure 2G to make this point more clear.

“Pulse-chase experiment showing that the stimulation of *intra*-complex T187 phosphorylation of p27 by tyrosine phosphorylation within the complex is greater than the stimulation of T187 phosphorylation taking place due to the displacement of unmodified p27 by pY74/pY88-p27 from the Cdk2/cyclin A complex.”

Minor points:

1) Figure 2B legend says that the experiments were performed in triplicates. It is better to label these lanes „triplicates“, otherwise it leaves the reader with the impression that the lanes have been left un-labeled by mistake, before the legend is read and guessed that they must be the triplicates.

We have added this label to the lanes.

2) The following sentence in the Supplementary Figure 2 legend is difficult to understand: “Error bars depict the standard deviation for the replicates at 1.0 and 2.0 μ M 1:1:1 complexes and Nus-Rb or p107”. Perhaps it should be “...and for incubations containing Nus-Rb or p107”?

We have modified the legend in Supplementary Figure 2 for clarity. This section now reads as follows:

“The results shown in Figure 2D depict the average fold stimulation of incorporation of 32 P into each substrate as a function of the phosphorylation status of p27. Error bars depict the standard deviation for the replicates at 1.0 and 2.0 μ M 1:1:1 complexes (p27/Cdk2/cyclin A) with Nus-Rb or p107 as the inter-molecular substrate. In each incubation the concentration of the intermolecular substrate (Nus-Rb or p107) matches that of the p27/Cdk2/cyclin A complex.”